# Discrete Diffusion Models: Novel Analysis and New Sampler Guarantees

**Yuchen Liang**[†], **Yingbin Liang**[†], **Lifeng Lai**[‡], **Ness Shroff**[†]

[†]The Ohio State University      [‡]University of California, Davis

## Abstract

Discrete diffusion models have recently gained significant prominence in applications involving natural language and graph data. A key factor influencing their effectiveness is the efficiency of discretized samplers. Among these, $\tau$-leaping samplers have become particularly popular due to their theoretical and empirical success. However, existing theoretical analyses of $\tau$-leaping often rely on somewhat restrictive and difficult-to-verify regularity assumptions, and their convergence bounds contain quadratic dependence on the vocabulary size. In this work, we introduce a new analytical approach for discrete diffusion models that removes the need for such assumptions. For the standard $\tau$-leaping method, we establish convergence guarantees in KL divergence that scale linearly with vocabulary size, improving upon prior results with quadratic dependence. Our approach is also more broadly applicable: it provides the first convergence guarantees for other widely used samplers, including the Euler method and Tweedie $\tau$-leaping. Central to our approach is a novel technique based on differential inequalities, offering a more flexible alternative to the traditional Girsanov change-of-measure methods. This technique may also be of independent interest for the analysis of other stochastic processes.

## 1 Introduction

Generative modeling is a core component of deep learning, aiming to generate samples that closely approximate distributions of training data. Recently, diffusion models [1, 2, 3] have gained significant attention. These models really work well in various generative tasks, particularly in image and video generation [4, 5]. Their effectiveness has been extensively documented in several comprehensive surveys [6, 7, 8].

*Discrete (i.e., discrete sample space)* diffusion models form a specialized subclass within the broader family of diffusion models and have gained increasing prominence in generative modeling. Similar to their continuous (i.e., continuous sample space) counterparts, they adopt the standard diffusion framework comprising of a forward and a reverse process. Differently, by operating over *discrete* sample spaces, both processes are formulated as discrete-state Markov chains, first under discrete-time [3] and later under continuous-time [9]. Since the seminal work [3], discrete diffusion models have been remarkably useful for a variety of discrete-data applications, achieving excellent performance in natural language processing (NLP) [10], graph generation [11, 12], and drug design [13]. Recent advances, such as SEDD and RADD, have further demonstrated language generation capabilities that rival those of traditional autoregressive models such as GPTs [10, 14].

Despite their empirical success, the theoretical understanding of discrete diffusion models remains limited. Existing theoretical studies on discrete-state diffusion models have mainly focused on various sampling methods. One focus has been on the *random*-step-size samplers, which sample both the next-state and the transition-times in the reverse process. This includes the Gillespie's algorithm [15] and the uniformization algorithm [16]. Specifically, the *uniformization* algorithm has been analyzed in [17, 18]. While these algorithms are able to simulate the reverse process exactly, their convergence

39th Conference on Neural Information Processing Systems (NeurIPS 2025).

guarantee is characterized only in a *stochastic* manner. In other words, there is no guarantee of fixed and finite number of iterations to achieve a target accuracy due to the algorithm's inherent randomness in the number of iterations, which, in the worst case, can be unbounded.

Another theoretical focus is on *deterministic*-step-size samplers, which enjoy finite-iteration guarantee for achieving a target accuracy. One type of such samplers is what we call as *Kolmogorov* sampler, which directly solves the piece-wise Kolmogorov equation at each discretized step, for which convergence guarantees were provided in [19, 20]. Such an approach may not be practically feasible, because it involves high computational costs in practice. At each step, the Kolmogorov sampler needs to solve a matrix exponential, which typically requires eigen-decomposition and multiplication of the reverse rate matrices (of size $S^d \times S^d$, where $S$ is the vocabulary size and $d$ is the data dimension). This renders the per-step computational complexity to be exponential in $d$.

A more practical deterministic-step-size sampler is $\tau$-*leaping* [21], which has drawn a lot of attention [9, 10, 18]. Rather than solving for the matrix exponential exactly, the algorithm applies all transitions within a single step simultaneously for each next-state and dimension. The existing theoretical studies [9, 18] on $\tau$-leaping have some fundamental limitations that need to be resolved. (1) **(Strong or hard-to-check assumptions)** Existing results require either strong assumptions on the estimation error or somewhat hard-to-check assumptions due to the analysis technique. Specifically, [9] assumes that the reverse rate matrix is well estimated under the $L^\infty$ error for each data sample and for each diffusion time, which is stronger compared to that only in expectation. [18] requires additional regularity assumptions on the diffusion path in order to invoke the Girsanov change-of-measure framework, which are usually hard to check in practice. (2) **(High dependence on vocabulary size)** Existing error bounds have high dependence on the vocabulary size $S$. In particular, the iteration complexity grows in fourth-power in both $d$ and $S$ for [9], and it grows quadratically in $S$ for [18].[1] This might be unsatisfactory in practice where $S$ is large (e.g., $S = 50257$ for GPT-2 tasks [10]). To this end, it is important to obtain a tighter bound on $S$. Thus, these open challenges can be summarized into the following intriguing question:

*Question 1: Can we establish convergence guarantees for $\tau$-leaping under more relaxed assumptions? Meanwhile, can we achieve a better dependency on $S$?*

While $\tau$-leaping is a practical sampler, it also has several weaknesses. For each step, the sampler requires sampling from a Poisson random variable *for each dimension and each token*, which becomes per-step sampling heavy especially when the vocabulary size $S$ is large. Also, especially for non-ordinal or categorical data, there could occur unmeaningful jumps such as multiple jumps within the same dimension or out-of-range jumps. For practical implementations, one usually needs additional constraints to restrict only one change to only one target location [9], which might further increase the sampling complexity. In comparison, empirical studies usually employ the Euler method or Tweedie $\tau$-leaping [10, 22], which are more sampling efficient than vanilla $\tau$-leaping. However, existing analytical tools are not directly applicable to these samplers.

*Question 2: Can we provide convergence guarantees for practically more efficient samplers having deterministic step-sizes, such as the Euler method and Tweedie $\tau$-leaping?*

This paper will provide affirmative answers to both of the above questions.

### 1.1   Our Contributions

Our main contribution in this paper lies in developing a novel analysis framework to analyze discrete diffusion models to improve/advance the current theory. Our detailed contributions are as follows.

**Novel Analysis Technique:** We develop a novel framework for analyzing discrete diffusion models. In particular, we directly analyze the rate-of-change of the KL-divergence between the true posterior and the sampling distribution, for which we provide an upper bound in terms of the respective rate matrices in the two processes by directly invoking the Kolmogorov equations. Our analysis (i) provides convergence guarantee without any regularity conditions, i.e., without requiring that the likelihood function on the sampling path is a local martingale, which is typically required for the Girsanov change-of-measure technique; (ii) enables to analyze a broader class of practical samplers, such as the Euler method and Tweedie $\tau$-leaping [10], for which it is challenging to apply the analysis based on the Girsanov change-of-measure framework.

---

[1]The error bound in [18] does not explicitly characterize the dependence on $S$. However, it is straightforward to derive the quadratic dependence on $S$ from their proof steps.

| Sampler | Space | Assump | Comp per-step | Sample per-step | Results: Num of steps | Reference |
|---|---|---|---|---|---|---|
| Uniform-ization | $\{0,1\}^d$ | int-path | $\mathcal{O}(d)$ | $\mathcal{O}(d)$ | $\mathrm{Pois}\left(\mathcal{O}\left(d\right)\right)$ | [17, Thm 6] |
| | $[S]^d$ | int-path | $\mathcal{O}(Sd)$ | $\mathcal{O}(d)$ | $\mathrm{Pois}\left(\mathcal{O}\left(dS\right)\right)$ | [18, Thm 4.9] |
| Kolmogo-rov | $[S]^d$ | int-path | $\mathcal{O}(S^{3d})$ | $\mathcal{O}(d)$ | $\mathcal{O}\left(\frac{\sqrt{dM}S}{\sqrt{\varepsilon}}\right)$ | [19, Cor 1] |
| DMPM | $\{0,1\}^d$ | disc-max | $\mathcal{O}(d)$ | $\mathcal{O}(1)$ | $\mathcal{O}\left(\frac{2^d d}{\varepsilon^2 \delta^d}\right)$ | [20, Cor 2.7] |
| $\tau$-leaping | $[S]^d$ | cont-max | $\mathcal{O}(Sd)$ | $\mathcal{O}(Sd)$ | $\mathcal{O}\left(\frac{d^4 S^4 + C d^2}{\sqrt{\varepsilon}}\right)$ | [9, Thm 1] |
| | $[S]^d$ | disc-sum | $\mathcal{O}(Sd)$ | $\mathcal{O}(Sd)^{**}$ | $\mathcal{O}\left(\frac{d^2 S^2}{\varepsilon}\right)$ | [18, Thm 4.7] |
| | $[S]^d$ | disc-sum | $\mathcal{O}(Sd)$ | $\mathcal{O}(Sd)$ | $\mathcal{O}\left(\frac{d^2 S}{\varepsilon}\right)$ | Thm 2 (here) |
| Euler | $[S]^d$ | disc-sum | $\mathcal{O}(Sd)$ | $\mathcal{O}(d)$ | $\mathcal{O}\left(\frac{d^2 S}{\varepsilon}\right)$ | Thm 3 (here) |
| Tweedie | $[S]^d$ | disc-sum | $\mathcal{O}(Sd)$ | $\mathcal{O}(d)$ | $\mathcal{O}\left(\frac{d^2 S}{\varepsilon}\right)$ | Thm 3 (here) |

Table 1: *Summary of results for discrete diffusion samplers in terms of the number of steps needed to achieve $\varepsilon$-accuracy in $\mathrm{KL}(q_\delta || p_{T-\delta})$ and the per-step computation and sampling complexity. Note that all log-dependencies are not shown. Here $d$ is the dimension (window-length for generative tasks), $S$ is the vocabulary size, $M$ is the upper bound of the score estimates (which grows in $\mathcal{O}(S)$), $\mathrm{Pois}(\lambda)$ is a Poisson r.v. with rate $\lambda$, and $q_\delta$ is such that $\mathrm{TV}(q_0, q_\delta) \lesssim d\delta$. **Comparison of results:** (i) Uniformization samplers suffer from random number of steps guarantee, whose actual iterations to convergence might grow unbounded. (ii) Kolmogorov samplers enjoy fixed and finite step guarantee, but suffer from exponential in $d$ per-step computational complexity, making it generally not practical. (iii) DMPM is an Euler-type method that differs from the standard Euler schemes [10, 22] studied in this paper. In particular, at most one coordinate is updated at each step, whereas the standard Euler sampler [10, 22] first constructs sampling probabilities for all coordinates and then performs a simultaneous categorical draw across all of them. After our initial submission of the paper, [20] provided an updated result, which is $\mathcal{O}(d/\varepsilon^2)$. (iv) The result on $\tau$-leaping in [9] has high dependence on $d$ and $S$ (note that $C = \Omega(S^2)$ is an implicit function of $S$), although is more efficient in $\epsilon$. The number of steps in [9] is calculated in order to reach $\sqrt{\varepsilon}$ total-variation error, which is weaker than the KL-divergence error considered in other guarantees in the table. The "cont-max" assumption is strong, which requires an upper bound for each time on each sample. The result on $\tau$-leaping in [18] has dependence on $S^2$. Note that $S$ is quite large for many NLP tasks. (v) Our result on $\tau$-leaping improves that of [18] by a factor of $S$. Our analysis also removes the regularity assumptions required in [18]. (vi) Our result on the Euler method and Tweedie $\tau$-leaping enjoys the same convergence rate as vanilla $\tau$-leaping, but these two samplers have smaller per-step sampling complexity than $\tau$-leaping by a factor of $S$.*

**Improved Result for $\tau$-leaping:** Based our analysis tools, we show that $\tau$-leaping generates a sample distribution that is close to the target distribution within an $\varepsilon$-KL accuracy with $\tilde{O}(d^2 S/\varepsilon)$ iteration steps. In particular, the iteration guarantee is *linear* in $S$ under the fully discrete score-entropy estimation loss, which improves the *quadratic* dependency on $S$ in [18]. This order-level improvement has important practical implications, especially because $S$ is often very large in NLP tasks (e.g., $S = 50257$ in [10]).

**New Convergence on Euler method, and Tweedie $\tau$-leaping:** Our analysis framework further provides convergence guarantees for practically efficient samplers: the Euler method and Tweedie $\tau$-leaping. The existing analysis tools do not seem to be applicable directly for lack of a path-wise measure defined for both samplers. Our result shows that these two samplers enjoy the same performance guarantee as $\tau$-leaping even with smaller sampling complexity at each step. Our approach involves constructing an approximate sampler which is asymptotically equivalent to both samplers, and then establishing the convergence guarantees for the constructed sampler. This constructed sampler might be of independent interest to future theoretical investigations of these two samplers.

## 1.2 Related Works

We have provided more prior works in Appendix A.

**Empirical Studies on Discrete Diffusion Models:** Unlike continuous-space diffusion models, discrete-space diffusion models are emerging as a strong contender in generative modeling, particularly for tasks involving discrete data [9, 10] (see some surveys in [12, 13]). The continuous-time discrete diffusion formulation was first developed in [9]. Recently, [10] first proposed the score-

entropy estimation error and achieved empirical success in text generation tasks. They also proposed a new discrete diffusion sampler by approximately solving the Tweedie's formula, which they call Tweedie $\tau$-leaping. For per-step sampling, note that most of these works use categorical sampling algorithms, yielding good empirical performances.

**Theory on Discrete Diffusion Models:** While there are numerous results for continuous-space diffusion models, the theoretical understanding of discrete diffusion models remains limited. Among them, [9] provided an early convergence analysis under the TV metric using $\tau$-leaping. However, the estimation error is quite strong, and the parameter dependencies are also high. More recently, under the score-entropy estimation errors, [17] provided the convergence result using the uniformization sampler on a $d$-dimensional hypercube, which was subsequently extended to general $[S]^d$ space in [18]. For deterministic-step-size samplers, [19, 20] performed analyses by assuming the accessibility of a perfect per-step sampler via solving the Kolmogorov equation, and [18] investigated the more practical $\tau$-leaping sampler. Among these works, [19] required the score-entropy loss to be evaluated on the continuous sampling path, whereas [18, 20] only required it to be on the discrete sampling grid. Notably, all of these works [17, 18, 19, 20] employed the Girsanov change-of-measure framework, which requires such regularity conditions (that the likelihood function is a path-wise local martingale) that are hard to check in practice.

## 2  Preliminaries of Discrete Diffusion Samplers

In this section, we provide the background of continuous-time discrete-space diffusion sampler.

### 2.1  The Forward Process

Let the initial (discrete) data $x_0 = \{x_0^1, \ldots, x_0^d\}$ consist of $d$ tokens, where each token $x_0^i \in [S]$ with $S$ being the cardinality of the token space. Hence, $x_0 \in [S]^d$. Let $q_0^i$ be the probability mass function (p.m.f.) of $x_0^i$, which is the probability simplex over $[0,1]^S$. We further let $q_0 \in [0,1]^{S^d}$ be entire p.m.f. of the initial data $x_0$.

The forward process can be characterized by a Continuous-Time Markov Chain (CTMC) from $t = 0$ to $t = T$, which is defined by an initial distribution $q_0$ and a *transition rate matrix* $R_t \in \mathbb{R}^{S^d \times S^d}$. Intuitively, each entry $R_t(x, y)$ in the rate matrix $R_t$ characterizes how fast state $x$ transitions to state $y$ at time $t$, where $x, y \in [S]^d$. Thus, for a sufficiently small time duration $\Delta t$, the conditional probability $q_{t+\Delta t|t}(y|x)$ of state $y$ at time $t + \Delta$ given state $x$ at time $t$ is given by

$$q_{t+\Delta t|t}(y|x) = \mathbb{1}\{y = x\} + R_t(x, y)\Delta t + o(\Delta t), \quad \forall x, y \in [S]^d. \tag{1}$$

Here $\mathbb{1}\{A\}$ is the indicator function of an event $A$. Equivalently, $q_t$ satisfies the Kolmogorov forward equation: $\frac{d}{dt} q_t(y) := \sum_{x \in [S]^d} q_t(x) R_t(x, y)$.

We now discuss several properties for the rate matrix $R_t$. For a valid CTMC, $R_t$ needs to satisfy that: 1) for all $x, y \in [S]^d$, $R_t(x, y) \geq 0$ if $x \neq y$; 2) $R_t(x, x) \leq 0$; and 3) $\sum_{y \in [S]^d} R_t(x, y) = 0$. Also, to make the computation tractable for large $S$ and $d$, a common practice is to make each token propagate *independently* and *homogeneously* (i.e., over the dimension) [9, 10]. Then, $R_t$ necessarily satisfies that, for all $x \neq y$, [9, Prop. 3]

$$R_t(x, y) = \begin{cases} R_t^{\text{tok}}(x^i, y^i) & \text{if Ham}(x, y) = 1, \\ 0 & \text{otherwise.} \end{cases} \tag{2}$$

Here $R_t^{\text{tok}} \in \mathbb{R}^{S \times S}$ is the token transition rate matrix (corresponding to $q_t^i$), and $\text{Ham}(x, y)$ denotes the number of unequal tokens between $x$ and $y$. We follow [9] and let $R_t^{\text{tok}} = \beta_t R_{\text{base}}$ for some noise schedule $\beta_t > 0$. In this paper, we are primarily interested in the constant noise schedule (i.e., $\beta_t \equiv 1$) as in the previous studies (e.g., [18, 19]). With such $R_t$, we can obtain an analytical solution for $q_{t|0}$ useful for training. Further, we primarily focus on the case where

$$R_{\text{base}} := \frac{1}{S} \mathbf{1}_S \mathbf{1}_S^\mathsf{T} - I_S,$$

which is common in many applications [3, 9, 10].[2] An immediate implication is that for all $x \in [S]^d$, $R_t(x, x) = -\sum_{y \neq x} R_t(x, y) = -\frac{S-1}{S} d$. Note that $q_T \approx \text{Uniform}([S]^d)$ for the chosen $R_{\text{base}}$.

---

[2]Most of our results can be extended for general $R_t$'s that satisfy [18, Assumption 4.3].

## 2.2  The Reverse Process

The forward process has a corresponding reverse process whose marginal distribution matches that of the forward process [23]. By [9, Prop. 1], such a reverse process is a CTMC with initial distribution $\breve{q}_0 := q_T$ and transition rate $\breve{R}_t$, where $\breve{R}_t$ satisfies

$$\breve{R}_t(x,y) := R_{T-t}(y,x)\frac{q_{T-t}(y)}{q_{T-t}(x)}, \quad \forall x \neq y, \quad \text{and} \quad \breve{R}_t(x,x) = -\sum_{y \neq x} \breve{R}_t(x,y). \tag{3}$$

We let the reverse process stop at $t = T - \delta$ with some small constant $\delta$. This technique is called early-stopping to prevent irregularities in the score when $t \to 0^+$. One immediate consequence is that with the $R_t$ in (2), whenever $\text{Ham}(x,y) \geq 2$, $\breve{R}_t(x,y) = 0$. Note that $\breve{q}_t := q_{T-t}$ for all $t \in [0, T - \delta]$.

## 2.3  The Sampling Process

In order to implement the reverse process, several approximations need to be made for sampling. Let $p_t$ be the marginal p.m.f. at time $t \in [0, T - \delta]$ in the sampling process. First, since $q_T$ is unavailable, we start the sampling process with $p_0 := \text{Uniform}([S]^d)$, which is the stationary distribution of the CTMC. Second, when $y \neq x$, we estimate the intractable ratio $\frac{q_t(y)}{q_t(x)}$ with $s_t(y,x)$, which is known as the *concrete score function*. Here we adopt the score-entropy (SE) loss [10] to train the score function, which is defined as

$$\mathcal{L}_{SE}(t; s_t) := \mathbb{E}_{x_t \sim q_t} \sum_{y \neq x_t} R_t(y, x_t) \left( s_t(y, x_t) - \frac{q_t(y)}{q_t(x_t)} - \frac{q_t(y)}{q_t(x_t)} \log \frac{s_t(y, x_t)}{q_t(y)/q_t(x_t)} \right). \tag{4}$$

Third, the continuous-path needs to be discretized for practical algorithms. Let $\{t_k\}_{k \in [N]}$ be the discretization points on which $s_{T-t_k}$ is accessible, where $t_0 = 0$ and $t_N = T - \delta$. Define the estimated rate on the sampling grid as

$$\hat{R}_{t_k}(x, y) := R_{T-t_k}(y, x) s_{T-t_k}(y, x). \tag{5}$$

$\tau$-**leaping:** A popular approximate sampler is called $\tau$-*leaping*, which has a deterministic number of sampling steps with polynomial per-step computation in $S$ and $d$. Given $x_{t_k}$, the next state is obtained as

$$x_{t_{k+1}} = x_{t_k} + \sum_{i=1}^{d} e^i \sum_{y^i \in [S]} (y^i - x_{t_k}^i) \text{Pois}\left( \hat{R}_{t_k}(x_{t_k}, x_{t_k}^{-i} \oplus_i y^i)(t_{k+1} - t_k) \right), \tag{6}$$

where $e^i$ is a unit (one-hot) vector on token $i \in [d]$, $\text{Pois}(\lambda)$ is a Poisson random variable with rate $\lambda$, and $v^{-i} \oplus_i a$ is a vector that replaces the $i$-th element of $v$ as $a \in [S]$. Intuitively, the sampler applies all transitions within $[t_k, t_{k+1})$ to a single component simultaneously, where the transition rate comes from the estimated reverse rate at the initial time. As shown in [9, Appendix B.5], the $\tau$-leaping process is equivalent to a CTMC with a piece-wise constant rate matrix given by

$$\hat{R}_t^{\tau\text{-leap}}(x, y) := \hat{R}_{t_k}(x_{t_k}, y - x + x_{t_k}), \quad \forall x \neq y, \ \forall t \in [t_k, t_{k+1}). \tag{7}$$

While $\tau$-leaping is popular in theoretical studies [9, 18], its sampling complexity is quite high because it requires drawing $\mathcal{O}(Sd)$ Poisson r.v.s at each sampling step. Rather, many empirical samplers, such as the Euler method and Tweedie $\tau$-leaping, draw only $\mathcal{O}(d)$ categorical r.v.s [10, 14, 22].

**Euler method:** The Euler method is given by, for each $k = 0, \ldots, N - 1$,

$$x_{t_{k+1}}^i = \begin{cases} a, & \text{w.p. } \hat{R}_k^i(x_{t_k}^i, a)(t_{k+1} - t_k), \ \forall a \neq x_{t_k}^i \\ x_{t_k}^i, & \text{w.p. } 1 + \hat{R}_k^i(x_{t_k}^i, x_{t_k}^i)(t_{k+1} - t_k) \end{cases}, \tag{8}$$

where $\hat{R}_k^i(z, a)$ is the token-wise rate defined as

$$\hat{R}_k^i(z, a) := \hat{R}_{t_k}(x_{t_k}, x_{t_k}^{-i} \oplus_i a) \mathbb{1}\left\{ z = x_{t_k}^i \right\}, \quad \forall a \neq x_{t_k}^i. \tag{9}$$

**Tweedie $\tau$-leaping:** The Tweedie $\tau$-leaping sampler is given by, for each $k = 0, \ldots, N - 1$,

$$x_{t_{k+1}}^i = \begin{cases} a, & \text{w.p. } \left( \left[ e^{-(t_{k+1}-t_k)R_{\text{base}}} \right]^{a:} s_{T-t_k}(x_{t_k}^{-i} \oplus_i \cdot, x_{t_k}) \right) \times \\ & \qquad \left[ e^{(t_{k+1}-t_k)R_{\text{base}}} \right]^{a, x_{t_k}^i}, \quad \forall a \neq x_{t_k}^i \\ x_{t_k}^i, & \text{otherwise} \end{cases}. \tag{10}$$

Note that neither sampler has such an $\hat{R}_t$ defined on the continuous-path $(t_k, t_{k+1})$. This renders the theoretical analysis for these two samplers to be still lacking, since existing analytical tools require the path-wise measure of the sampling process to be well-defined (for all $t \in [0, T - \delta]$).

### 2.4 Key Notations

Let $x^i(1 \leq i \leq d)$ denote the $i$-th element of a vector $x \in [S]^d$ and $x^{-i} \in [S]^{d-1}$ denote the $i$-th element removed. Define $\mathrm{Ham}\,(x, y)$ as the Hamming distance between two vectors $x$ and $y$. For a positive integer $n$, $[n] := \{1, \ldots, n\}$. Write $\mathbf{1}_S$ as a vector of length $S$ that contains all 1's, and $I_S$ as an identity matrix of size $S \times S$. See a full list of notations in Appendix B.

## 3 Main Convergence Results

The $\tau$-leaping sampler has been analyzed in [9, 18] with convergence guarantees. However, the convergence rate in [9] has high dependence on $S$ and $d$ (which grows in fourth-power in both $d$ and $S$). Further, [9] assumes that the reverse rate matrix is well estimated under $L^\infty$ error. The study in [18] relaxed such an assumption to be in expectation, and the convergence rate in [18] grows only quadratically in $S$. However, the study in [18] requires additional difficult-to-check assumptions on the diffusion path in order to invoke the Girsanov change-of-measure framework.

In this section, we develop a new analytical framework, which removes the need for the regularity assumptions used in [18], and to further improve their convergence rate with a lower-order dependence on $S$.

### 3.1 Convergence Guarantees for General Sampling Processes

We first propose the following KL-divergence decomposition for general reverse rates, without any regularity assumptions for the sampling path. Indeed, the result is applicable to any CTMC with general forward rate $R_t$ as long as the reverse rate $\breve{R}_t$ is well-defined [23, Chapter 1].

**Theorem 1.** *Recall the true reverse process with rate defined in* (3). *Suppose that $p_t$ also follows a CTMC, with initial distribution $p_0$ and rate $\hat{R}_t$. Then,*

$$
\mathrm{KL}(\breve{q}_{T-\delta}||p_{T-\delta}) \leq \mathrm{KL}(\breve{q}_0||p_0) + \sum_{k=0}^{N-1} \mathbb{E}_{x_{t_k} \sim \breve{q}_{t_k}} \left[ \mathrm{KL}(\breve{q}_{t_{k+1}|t_k}(\cdot|x_{t_k})||p_{t_{k+1}|t_k}(\cdot|x_{t_k})) \right]
$$

$$
\leq \mathrm{KL}(\breve{q}_0||p_0) + \sum_{k=0}^{N-1} \int_{t_k}^{t_{k+1}} \mathbb{E}_{x_t \sim \breve{q}_t} \left[ \sum_{y \neq x_t} \hat{R}_t(x_t, y) - \breve{R}_t(x_t, y) + \breve{R}_t(x_t, y) \log \frac{\breve{R}_t(x_t, y)}{\hat{R}_t(x_t, y)} \right] \mathrm{d}t.
$$

$$
(11)
$$

Theorem 1 indicates that, *without any regularity condition*, the final KL-divergence between the marginal distributions of the true and the mismatched process can be upper-bounded by the sum of (1) the divergence between the initial distribution and (2) that accumulated along the sampling path due to mismatched rate matrices. Also, different from continuous diffusion models [24], the rate of accumulation at each time $t$ is characterized not by Fisher divergence but by Bregman divergence generated by the negative entropy function.

The proof is provided in Appendix E. Our proof of Theorem 1 takes a differential inequality argument, which is different from the approaches used in existing works. After invoking the chain-rule of the KL divergence (cf. (24)), the key is to provide an upper bound for the rate-of-change of the KL-divergence between the true posterior and the sampling distribution. We then convert it into the rate-of-change of the respective probabilities themselves, which can be further characterized by the corresponding CTMC rate matrices by invoking the Kolmogorov equation. Rearranging the terms, we finally obtain an upper bound in terms of only the rate matrices.

While our idea comes from [24, Lemma 6 and Proposition 8] (which studied continuous diffusion models), there are several key differences: (1) In contrast to the continuous diffusion studied in [24], there is no Fokker-Planck equation defined for discrete diffusion (since the space is discrete). Instead, we need to use the Kolmogorov equation specifically tailored to a CTMC, which is related to the special CTMC rate matrix. (2) Because of this, we need to characterize the reverse rate of a CTMC not in terms of the marginal probabilities (as in [9]) but ones that are *conditioned on future observations*. This is a non-trivial extension, whose proof is given in Lemma 9.

**Comparison with Girsanov-based approaches:** Our differential-inequality based approach for proving Theorem 1 is more advantageous than previous Girsanov-based approaches (see [18, Corollary 3.4], [19, Lemma 1], and [20, Theorem F.3]), in two ways. (1) First, we do not require the regularities conditions necessary to apply the Girsanov change-of-measure (see [18, Remark A.12],

[19, Theorem 3], and [20, Theorem F.3]). (2) [18, 19, 20] assume a particular parameterization of the sampling rate as $\hat{R}_t(x, y) = R_{T-t}(y, x)s_{T-t}(y, x)$, whereas our analysis allows $\hat{R}_t(x, y)$ to be generic. This might be useful, for example, when the rate is not obtained through minimizing the score-entropy, or when there is general score mismatch that arises from a mismatched target [25]. Nonetheless, Theorem 1 can be applied directly to the aforementioned particular parameterization $\hat{R}_t(x, y)$ to obtain the following Corollary 1, where the score-entropy is used. The proof is straightforward, which is provided for completeness in Appendix F.

**Corollary 1.** *Under the parameterization that* $\hat{R}_t(x, y) = R_{T-t}(y, x)s_{T-t}(y, x)$, *we have*

$$\mathrm{KL}(\tilde{q}_{T-\delta}||p_{T-\delta}) \leq \mathrm{KL}(\tilde{q}_0||p_0) + \sum_{k=0}^{N-1} \int_{t_k}^{t_{k+1}} \mathcal{L}_{SE}(T-t; s_{T-t})\mathrm{d}t. \tag{12}$$

*Here* $\mathcal{L}_{SE}$ *is the score-entropy defined in* (4).

### 3.2 Improved Parameter Dependence for $\tau$-leaping

In this subsection, we characterize the convergence rate of $\tau$-leaping with explicit dependencies on $d$ and $S$. To this end, we employ the following standard assumptions.

**Assumption 1** (Score Estimation Error). Recall $\mathcal{L}_{SE}$ as defined in (4). The score estimation satisfies

$$\sum_{k=0}^{N-1} (t_{k+1} - t_k)\mathcal{L}_{SE}(T - t_k; s_{T-t_k}) \leq \varepsilon_{\mathrm{score}}. \tag{13}$$

Note that Assumption 1 captures the error that $s_{t_k}$ incurs for estimating the score function in terms of the loss value at $T - t_k$'s. We have provided a table to compare different assumptions for score estimation in Table 1. In particular, for those works using score-entropy estimation errors, [17, Assumption 1] and [19, Assumption 1] require that $s_t$ (or $s_{t_k}$) is well-estimated along the integral-path of the sampling process. This assumption is typically hard to verify in practice because of the continuity of the sampling path. [20, Cor 2.7] requires that the maximum error over the discrete sampling grid is well-controlled, which is stronger than our Assumption 1. Our Assumption 1 is the same as [18, Assumption 4.6], which assumes that the sum-averaged error over the discrete grid is well-controlled, which can be practically verified.

**Assumption 2** (Bounded Score Estimates). The score estimates $s_{t_k}$'s satisfy $s_{t_k}(x, y) \in [M^{-1}, M]$ for all $x, y \in [S]^d$ and $k = 0, \ldots, N - 1$.

Assumption 2 is commonly adopted in the previous studies such as in [18, 19]. In practice, this can be satisfied with score-clipping during training [19]. Indeed, as shown in Theorem 2, the convergence error bounds depend only on $\log M$.

We are now ready to present our main result below.

**Theorem 2.** *Suppose that Assumptions 1 and 2 hold. Using the $\tau$-leaping sampler, and choosing* $t_{k+1} - t_k \leq \kappa \min \{1, T - t_k\}$, *we have*

$$\mathrm{KL}(q_\delta||p_{T-\delta}) \lesssim d(\log S)e^{-T} + \varepsilon_{score} + \kappa d^2 S(T + \log(MS\delta^{-1})), \tag{14}$$

*where $q_\delta$ satisfies* $\mathrm{TV}(q_0, q_\delta) \lesssim d\delta$.

*Furthermore, by letting $t_{k+1} - t_k = \kappa \min \{1, T - t_k\}$ and choosing $T = \log(d(\log S)/\varepsilon)$, we have that $\mathrm{KL}(q_\delta||p_{T-\delta})$ achieves $\varepsilon$ error with $N = \tilde{O}\left(d^2S/\varepsilon\right)$ sampling steps.*

Theorem 2 indicates that it takes at most $\tilde{O}(d^2S/\varepsilon)$ iterations to approximate a $\delta$-perturbed distribution of $q_0$ to $\varepsilon$-accuracy in KL-divergence. The *linear* dependence on $S$ improves upon the best previously known result for $\tau$-leaping in [18] by a factor of $\mathcal{O}(S)$.[3] This order-level improvement has important practical implications, because $S$ is often very large for many NLP tasks (e.g., $S = 50257$ [10]). We have performed a numerical study to validate such linear dependence in Figure 1 in Appendix D. The full proof of Theorem 2 is in Appendix G, and we have provided a proof sketch in Section 4.

---

[3]The error bound in [18] does not explicitly characterize the dependence on $S$. However, it is straightforward to derive the quadratic dependence on $S$ from their proof steps.

### 3.3 Convergence Guarantees for Euler Method and Tweedie $\tau$-leaping

A significant obstacle in establishing the guarantees for the Euler method and Tweedie $\tau$-leaping is that both samplers are defined directly on the discrete sampling grids and thus lack an intermediate rate function. Our approach to tackle this challenge includes the following steps. (i) First, we construct a non-trivial approximate sampler with explicit intermediate rate that allows for *categorical* per-step sampling. (ii) We then show that our construction is asymptotically equivalent (in the categorical sampling probabilities) to both the Euler method and Tweedie $\tau$-leaping. (iii) Then, in order to establish convergence guarantees for these samplers, we show that asymptotically equivalent samplers would also result in the same asymptotic rate in KL-divergence. In particular, this step is based on our step-wise KL-divergence decomposition in (26). In comparison, the Girsanov change-of-measure technique cannot be applied here directly due to the lack of a path-wise measure defined for both samplers. (iv) Finally, we show that our constructed approximate sampler enjoys the same rate as vanilla $\tau$-leaping does. The full proof is in Appendix H.

**Theorem 3.** *Suppose that Assumptions 1 and 2 hold. For both the Euler method and Tweedie $\tau$-leaping, choosing $t_{k+1} - t_k \leq \kappa \min\{1, T - t_k\}$, we have*

$$\mathrm{KL}(q_\delta \| p_{T-\delta}) \lesssim d(\log S)e^{-T} + \varepsilon_{score} + \kappa d^2 S(T + \log(MS\delta^{-1})). \tag{15}$$

*Here* $\mathrm{TV}(q_0, q_\delta) \lesssim d\delta$. *Similarly, if we take $t_{k+1} - t_k = \kappa \min\{1, T - t_k\}$, it suffices that $T = \log(d(\log S)/\varepsilon)$ and $N = \tilde{O}\left(d^2 S/\varepsilon\right)$ to reach $\varepsilon$ KL-divergence error.*

Notably, this is the *first* theoretical convergence guarantee characterized for the Euler method and Tweedie $\tau$-leaping. Compared with vanilla $\tau$-leaping, since both samplers require the same number of iterations but a decreased number of samples per-step (by a factor of $\mathcal{O}(S)$), our Theorem 3 shows that the Euler method and Tweedie $\tau$-leaping enjoy less overall sampling complexity for a given target accuracy $\varepsilon$. This benefit becomes more significant when $S$ is large, as in many practical tasks [10]. We have also numerically compared these samplers in Figure 2 in Appendix D.

## 4 Proof Sketch of Theorem 2

In this section, we provide a proof sketch of Theorem 2 to describe the main idea of our analysis approach. The full proof is in Appendix G. Upon invoking the KL-divergence decomposition in Theorem 1, we can decompose the total error into three different errors, where the discretization error is the rate-determining term. For the two terms of the discretization error, we further identify one of the dominant term and provide an error upper bound directly in expectation.

**Comparison with the approach of [18]:** Our proof is different from [18] in the following ways. (i) In Step 1, we do not use the Girsanov change-of-measure framework to start with, thus eliminating the need for any regularity conditions, which restrict path-wise integrability and are typically hard to check in practice (see [18, Corollary 3.4 and Remark A.12]). Rather, our Theorem 1 provides a more general starting point for which no such regularity conditions are needed. (ii) In Step 2, for determining parameter dependency, we do not construct a stochastic-integral framework where Ito's Lemma is needed for the analysis of discretization error (see [18, Theorem A.10 and Proposition C.4]). Instead, we directly identify the dominant error term by invoking the Kolmogorov equation, thus eliminating the need of such stochastic-integral formulation in the analysis. (iii) In Step 3, we do not employ a uniform upper bound (in $x$ and $y$) for the score difference to control the discretization error as in [18, Proposition C.2]. Differently, our upper bound is only in expectation of $x_t$, which enables us to reduce the quadratic dependency on $S$ to linear dependency.

In the following, we divide the proof into three steps.

**Step 1: Decomposing total error (Theorem 1).** Following Theorem 1, we can decompose the total error as

$$\mathrm{KL}(\breve{q}_T \| p_T) \leq \underbrace{\mathrm{KL}(\breve{q}_0 \| p_0)}_{\text{initialization error}} + \underbrace{\sum_{k=0}^{N-1}(t_{k+1} - t_k)\mathbb{E}_{x_{t_k} \sim \bar{q}_{t_k}}\left[g_{t_k}(x_{t_k})\right]}_{\text{estimation error}} +$$

$$\underbrace{\sum_{k=0}^{N-1}\int_{t_k}^{t_{k+1}}\mathbb{E}_{\substack{x_t \sim \bar{q}_t \\ x_{t_k} \sim \bar{q}_{t_k}}}\left[g_t(x_t) - g_t(x_{t_k})\right] + \mathbb{E}_{x_{t_k} \sim \bar{q}_{t_k}}\left[g_t(x_{t_k}) - g_{t_k}(x_{t_k})\right] \mathrm{d}t,}_{\text{discretization error}} \tag{16}$$

where we have defined $g_t(x_t) := \sum_{y \neq x_t} \hat{R}_t(x_t, y) - \breve{R}_t(x_t, y) + \breve{R}_t(x_t, y) \log \frac{\breve{R}_t(x_t, y)}{\hat{R}_t(x_t, y)}$. Indeed $g_t$ is a Bregman divergence generated by the negative entropy function. Here from [19, Proposition 2] and [18, Theorem C.1], the initialization error satisfies $\mathrm{KL}(\bar{q}_0 \| p_0) \lesssim (d \log S) e^{-T}$. Also, the estimation error satisfies

$$\sum_{k=0}^{N-1} (t_{k+1} - t_k) \mathbb{E}_{x_{t_k} \sim \bar{q}_{t_k}} [g_{t_k}(x_{t_k})] = \sum_{k=0}^{N-1} (t_{k+1} - t_k) \mathcal{L}_{SE}(T - t_k; s_{t_k}) \leq \varepsilon_{\mathrm{score}}. \quad (17)$$

It remains to provide an upper-bound for the two terms, which constitute the discretization error.

**Step 2: Identifying dominant term for discretization error (Lemma 1).** As shown above, the discretization error consists of two terms: one for the time-difference in the argument of $g_t$ (in expected value), and the other for the difference in $g_t$ itself. For the former term, the expected difference in the argument can be upper-bounded using the Kolmogorov forward equation and the rate properties. Indeed, we can show that the former term is decaying faster than the other, which further implies that it does not contribute to the total error:

$$\int_0^{T-\delta} \mathbb{E}_{\substack{x_t \sim \bar{q}_t \\ x_{t_k} \sim \bar{q}_{t_k}}} [g_t(x_t) - g_t(x_{t_k})] = \kappa \cdot O \left( \varepsilon_{\mathrm{score}} + \int_0^{T-\delta} \mathbb{E}_{x_{t_k} \sim \bar{q}_{t_k}} [g_t(x_{t_k}) - g_{t_k}(x_{t_k})] \, \mathrm{d}t \right)$$

$$= o(\kappa). \quad (18)$$

**Step 3: Bounding dominant term for discretization error (Lemmas 3 and 5 and Equation (29)).** Now, we control the latter term in the discretization error, which is also the dominant-rate error term. We first upper-bound this term as an expected difference of the reverse CTMC rate matrix (Lemma 3 and Equation (29)):

$$\mathbb{E}_{x_{t_k} \sim \bar{q}_{t_k}} (g_t(x_{t_k}) - g_{t_k}(x_{t_k})) \lesssim (1 + \log(MS\delta^{-1})) \mathbb{E}_{x_{t_k} \sim \bar{q}_{t_k}} \sum_{y \neq x_{t_k}} \left| \breve{R}_t(x_{t_k}, y) - \breve{R}_{t_k}(x_{t_k}, y) \right|$$

$$\lesssim (1 + \log(MS\delta^{-1})) \mathbb{E}_{x_{t_k} \sim \bar{q}_{t_k}} \sum_{\substack{y \neq x_{t_k} \\ \mathrm{Ham}(y, x_{t_k}) = 1}} \left| \frac{q_{T-t_k}(y)}{q_{T-t_k}(x_{t_k})} - \frac{q_{T-t}(y)}{q_{T-t}(x_{t_k})} \right| R_{T-t_k}(y, x_{t_k}). \quad (19)$$

Thus, it is essential to deal with the time difference of the likelihood ratios (i.e., concrete scores). One common way is to exploit the continuity of this ratio and to upper-bound its derivative for every fixed $x$ and $y$ such that $\mathrm{Ham}(x, y) = 1$. Indeed, this is the approach taken by [18, Prop. C.2], which will result in an $\mathcal{O}(S^2)$ dependency, as we show in Lemma 4 (for purpose of comparison). Instead, our approach here directly provides an upper bound in expectation, which enables us to reduce a factor of $\mathcal{O}(S)$ (see Lemma 5). With this improved bound, the discretization error $\mathcal{W}_{\mathrm{disc}}$ satisfies that

$$\mathcal{W}_{\mathrm{disc}} \lesssim \sum_{k=0}^{N-1} d^2 S \max\{1, (T - t_{k+1})^{-2}\}(t_{k+1} - t_k)^2. \quad (20)$$

Note that this results in a tighter upper bound with linear $S$ dependency.

Finally, combining the steps above, and invoking [24, Lemma 18], we can determine the overall parameter dependencies in the above summation, which shows that

$$\sum_{k=0}^{N-1} \max\{1, (T - t_{k+1})^{-2}\}(t_{k+1} - t_k)^2 \lesssim \kappa(T + \log \delta^{-1}). \quad (21)$$

Also, from the last part of [17, Theorem 6], the perturbation due to early-stopping satisfies that

$$\mathrm{TV}(q_0, q_\delta) \lesssim d\delta, \quad \text{as } \delta \to 0. \quad (22)$$

## 5 Conclusion

In this paper, we have introduced a new analytical approach for discrete diffusion models that removes the need for any regularity assumptions required in the previous Girsanov change-of-measure techniques. For the standard $\tau$-leaping sampler, we have established convergence guarantees that scale linearly with the vocabulary size, improving upon prior results with quadratic dependence. We have also provided the first convergence guarantees for other widely used samplers, including the Euler method and Tweedie $\tau$-leaping. In the future, it might be interesting to investigate acceleration techniques that further reduce the order of $d$ and $S$ dependence.

## Acknowledgments and Disclosure of Funding

The work of Y. Liang, Y. Liang and N. Shroff was supported in part by the U.S. National Science Foundation under the grants: NSF AI Institute (AI-EDGE) 2112471, ECCS-2413528, CNS-2312836, CNS-2223452, CNS-2225561, and was sponsored by the Army Research Laboratory under Cooperative Agreement Number W911NF-23-2-0225. The work of L. Lai was supported in part by the U.S. National Science Foundation under the grants: CCF-2232907 and ECCS-2448268. The views and conclusions contained in this document are those of the authors and should not be interpreted as representing the official policies, either expressed or implied, of the Army Research Laboratory or the U.S. Government. The U.S. Government is authorized to reproduce and distribute reprints for Government purposes notwithstanding any copyright notation herein.

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

# Appendix

## A  Related Works

**Theory on Continuous Diffusion Models:** There have been many works that have explored the performance guarantees of continuous-space diffusion models. While initial studies focused on $L^\infty$ score estimation error and exponential error bounds [26, 27], subsequent studies developed polynomial error bounds under $L^2$ score estimation error (e.g., [28, 29, 30, 31, 32]). In particular, [29] first employed the Girsanov change-of-measure framework for continuous diffusion models and obtained guarantees for Lipschitz-score distributions. This result was later improved under a differential-inequality-based analysis in [24], which removed all regularity conditions and enlarged the distributional set to enforce the Lipschitz condition only for the target and to include all finite-variance targets. Their idea inspired our analysis to investigate a differential-inequality based analysis

also for the discrete diffusion models. Apart from the works mentioned above, there are other works that provide convergence guarantees on the discrete-time formulation directly [33, 34, 35], on the deterministic sampler [36, 37, 38, 39], on the Wasserstein distance [40, 41], and on the modified predictor-corrector sampling method [42]. Recently, [43] provided a unified reverse-transition-kernel framework that include these stochastic samplers for continuous-space diffusion models.

**Empirical Works on Discrete Diffusion Models:** Different from continuous-space diffusion models, discrete-space diffusion models are rising as a strong candidate in generative modeling, especially for those tasks involving discrete data such as texts [10], images [9], and graphs [12], having applications even in the biochemical field [13]. Early ideas of discrete diffusion models can be traced back to [1], which was subsequently extended in [44, 3]. The continuous-time counterpart using CTMC was developed in [9]. In particular, all models in [44, 3, 9] are trained according to a variational inference objective, which maximizes an Evidence Lower-Bound (ELBO). While empirically effective, it is hard for one to characterize the estimation error, which is implicit in the Jensen bound. Recently, [10] first proposed the score-entropy estimation error which is easy to train, and they achieved empirical success compared to autogressive models in text generation tasks. They also proposed a new discrete diffusion sampler by approximately solving the Tweedie's formula, which they call the Tweedie $\tau$-leaping sampler. Other than an improved training objective, there are other works that are focused on the conditional guidance in discrete diffusion models [11, 22, 45], on the discrete flow models [46, 47], on the non-Markovian sampling process [48], on fine-tuning [49], to name a few. For per-step sampling methods, note that the majority of these works (only except [9]) use categorical sampling methods, yielding good empirical performances.

**Theory on Discrete Diffusion Models:** While there have been flourishing results for continuous-space diffusion models, the theoretical understanding of discrete diffusion models remains limited. All convergence results are given in Table 1. Among them, [9] provided an early convergence analysis under the TV metric using the $\tau$-leaping sampler. However, the estimation error is quite strong, and the parameter dependencies are also high. More recently, under the score-entropy estimation errors, [17] provided the convergence result using the uniformization sampler on a $d$-dimensional hypercube, which was subsequently extended to general $[S]^d$ space in [18]. For deterministic-step-size samplers, [19] performed analyses by assuming the accessibility of a perfect per-step sampler via solving the Kolmogorov equation, [20] analyzed an Euler-type method which differs from the standard Euler schemes [10, 22],[4] and [18] investigated the more practical $\tau$-leaping sampler. Among these works, [19] required that the score-entropy loss is evaluated on the continuous sampling path, whereas [18, 20] only required such loss to be evaluated on the discrete sampling grid. Notably, all of these works [17, 18, 19, 20] employed the Girsanov change-of-measure framework, which requires such regularity conditions (that the likelihood function is a path-wise local martingale) that are hard to check in practice. Recently, [50] investigated possible acceleration schemes in discrete diffusion models, and [51] used discrete diffusion techniques to solve for continuous diffusion problems under quantization.

## B  Full List of Notations

For any two functions $f(d, \delta, \varepsilon)$ and $g(d, \delta, \varepsilon)$, we write $f(d, \delta, \varepsilon) \lesssim g(d, \delta, \varepsilon)$ (resp. $f(d, \delta, \varepsilon) \gtrsim g(d, \delta, \varepsilon)$) for some universal constant (not depending on $\delta$, $d$ or $T$) $L < \infty$ (resp. $L > 0$) if $\limsup_{\varepsilon \to 0} |f(d, \delta, \varepsilon)/\ g(d, \delta, \varepsilon)| \leq L$ (resp. $\liminf_{\varepsilon \to 0} |f(d, \delta, \varepsilon)/g(d, \delta, \varepsilon)| \geq L$). We write $f(d, \delta, \varepsilon) \asymp g(d, \delta, \varepsilon)$ when both $f(d, \delta, \varepsilon) \lesssim g(d, \delta, \varepsilon)$ and $f(d, \delta, \varepsilon) \gtrsim g(d, \delta, \varepsilon)$ hold. Unless otherwise specified, we write $x^i (1 \leq i \leq d)$ as the $i$-th element of a vector $x \in [S]^d$ and $[A]^{ij}$ as the $(i, j)$-th element of a matrix $A$. Also, write $x^{-i} \in [S]^{d-1}$ as the $i$-th element removed. Define $\mathrm{Ham}(x, y)$ as the Hamming distance between two vectors $x$ and $y$ (which is equal to the number of non-equal elements). We also write $\mathrm{Ham}(x) = \mathrm{Ham}(x, 0)$ for brevity. For matrices $A, B$, $\mathrm{Tr}(A)$ is the trace of $A$, and $A \preceq B$ means that $B - A$ is positive semi-definite. For a positive integer $n$, $[n] := \{1, \ldots, n\}$. Write $\mathbf{1}_S$ as a vector of length $S$ that contains all 1's, and $I_S$ as an identity matrix of size $S \times S$. Write $\mathbb{1}\{A\}$ as the indicator function of an event $A$.

---

[4]In the DMPM algorithm in [20], at most one coordinate is updated at each step, whereas the standard Euler sampler [10, 22] first constructs sampling probabilities for all coordinates and then performs a simultaneous categorical draw across all of them. Consequently, the number of flips at each step is a random variable taking values in 0 to $d$ (where $d$ is the dimension).

## C  Implementations of Discrete Diffusion Samplers

In this section, we provide the implementation of two practical sampling algorithms typically used in empirical studies, namely the Euler method and Tweedie $\tau$-leaping [10]. We also provide our construction of an approximate discrete sampler for the proof of Theorem 3, which we name as Truncated $\tau$-leaping.

---

**Algorithm 1:** Euler method (e.g., in [10, 22])

---

**Input:** initial sample $x_{t_0} \sim p_0$, discretization points $\{t_k\}_{k=0}^N$ (with $t_0 = 0$ and $t_N = T - \delta$), estimated score on these discretized points $s_{T-t_k}$

1  **for** $k = 0$ **to** $N - 1$ **do**
2     $\hat{R}_{t_k}(x, y) \leftarrow R_{T-t_k}(y, x) s_{T-t_k}(y, x)$;
3     **for** $i = 1$ **to** $d$ **do**
4        Recall $\hat{R}_k^i(z, a)$ from (9). Draw $x_{t_{k+1}}^i$ as follows:

$$
x_{t_{k+1}}^i = \begin{cases} a\ (\neq x_{t_k}^i), & \text{w.p. } \hat{R}_k^i(x_{t_k}^i, a)(t_{k+1} - t_k) \\ x_{t_k}^i, & \text{w.p. } 1 + \hat{R}_k^i(x_{t_k}^i, x_{t_k}^i)(t_{k+1} - t_k) \end{cases}
$$

5     **end**
6  **end**
   **Return:** $x_{t_N}$

---

---

**Algorithm 2:** Tweedie $\tau$-leaping [10]

---

**Input:** initial sample $x_{t_0} \sim p_0$, discretization points $\{t_k\}_{k=0}^N$ (with $t_0 = 0$ and $t_N = T - \delta$), estimated score on these discretized points $s_{T-t_k}$

1  **for** $k = 0$ **to** $N - 1$ **do**
2     **for** $i = 1$ **to** $d$ **do**
3        Draw $x_{t_{k+1}}^i$ as follows:

$$
x_{t_{k+1}}^i = \begin{cases} a\ (\neq x_{t_k}^i), & \text{w.p. } \left( \left[ e^{-(\bar{\beta}_{T-t_k} - \bar{\beta}_{T-t_{k+1}})R_{\text{base}}} \right]^{a:} s_{T-t_k}\left( x_{t_k}^{-i} \oplus_i \cdot, x_{t_k} \right) \right) \times \\ & \qquad\qquad \left[ e^{(\bar{\beta}_{T-t_k} - \bar{\beta}_{T-t_{k+1}})R_{\text{base}}} \right]^{a, x_{t_k}^i} \\ x_{t_k}^i, & \text{otherwise} \end{cases}
$$

4     **end**
5  **end**
   **Return:** $x_{t_N}$

---

## D  Numerical Simulations

In this section, we provide some numerical simulations to validate our theoretical results. The target distribution is a synthetic autoregressive model with given coefficients.

## E  Proof of Theorem 1

Write $\mathcal{X} := [S]^d$. To begin, we note the chain-rule of KL divergence as [35, Theorem 1] (cf. [24, Theorem 1])

$$
\text{KL}(\breve{q}_T || p_T) \leq \text{KL}(\breve{q}_0 || p_0) + \sum_{k=0}^{N-1} \mathbb{E}_{x_{t_k} \sim \breve{q}_k} \left[ \text{KL}(\breve{q}_{t_{k+1}|t_k}(\cdot | x_{t_k}) || p_{t_{k+1}|t_k}(\cdot | x_{t_k})) \right], \tag{24}
$$

where, for each $x_{t_k} \in \mathcal{X}$,

$$
\text{KL}(\breve{q}_{t_{k+1}|t_k}(\cdot | x_{t_k}) || p_{t_{k+1}|t_k}(\cdot | x_{t_k})) = \text{KL}(\breve{q}_{s|t_k}(\cdot | x_{t_k}) || p_{s|t_k}(\cdot | x_{t_k})) +
$$

---

**Algorithm 3:** Truncated $\tau$-leaping (used in Appendix H)

---

**Input:** initial sample $x_{t_0} \sim p_0$, discretization points $\{t_k\}_{k=0}^N$ (with $t_0 = 0$ and $t_N = T - \delta$), estimated score on these discretized points $s_{T-t_k}$

**1 for** $k = 0$ **to** $N - 1$ **do**

**2**     $\hat{R}_{t_k}(x, y) \leftarrow R_{T-t_k}(y, x)s_{T-t_k}(y, x)$;

**3**     **for** $i = 1$ **to** $d$ **do**

**4**        For each $z, a \in [S]$, recall that the token-wise rate $\hat{R}_k^i \in \mathbb{R}^{S \times S}$ is defined in (9) as

$$\hat{R}_k^i(z, a) := \hat{R}_{t_k}(x_{t_k}, x_{t_k}^{-i} \oplus_i a)\mathbb{1}\left\{z = x_{t_k}^i\right\}, \quad \forall a \neq x_{t_k}^i$$

       with $\hat{R}_k^i(x_{t_k}^i, x_{t_k}^i) := -\sum_{\substack{a \in [S] \\ a \neq x_{t_k}^i}} \hat{R}_k^i(x_{t_k}^i, a)$;

**5**        Draw $x_{t_{k+1}}^i$ as follows:

$$x_{t_{k+1}}^i = \begin{cases} a\ (\neq x_{t_k}^i), & \text{w.p. } \frac{\hat{R}_k^i(x_{t_k}^i, a)}{-\hat{R}_k^i(x_{t_k}^i, x_{t_k}^i)}\left(1 - \exp\left(\hat{R}_k^i(x_{t_k}^i, x_{t_k}^i)(t_{k+1} - t_k)\right)\right) \\ x_{t_k}^i, & \text{w.p. } \exp\left(\hat{R}_k^i(x_{t_k}^i, x_{t_k}^i)(t_{k+1} - t_k)\right) \end{cases}$$

$$\tag{23}$$

**6**     **end**

**7 end**

**Return:** $x_{t_N}$

---

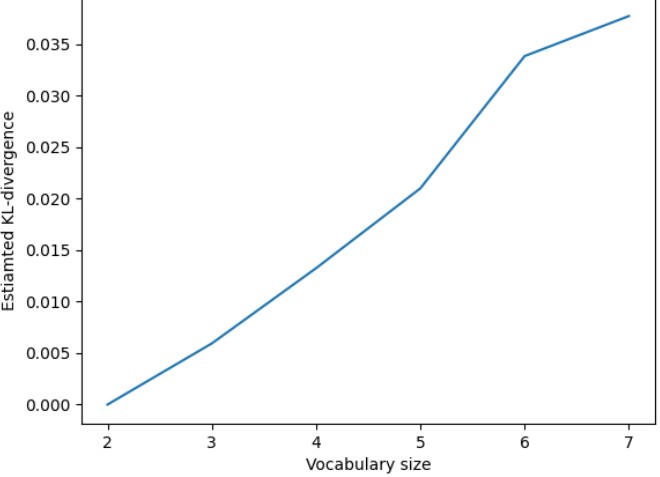

Figure 1: Estimated KL-divergence between the target and the sampling distribution. The target is generated autoregressively over the dimensions. Here $d = 2$. We use Euler method to obtain 2000000 samples to estimate the KL divergence.

$$\int_s^{t_{k+1}} \frac{\partial}{\partial t} \text{KL}(\bar{q}_{t|t_k}(\cdot|x_{t_k})||p_{t|t_k}(\cdot|x_{t_k}))\mathrm{d}t.$$

By Lemma 8, we can take the limit $s \downarrow t_k$ which yields

$$\text{KL}(\bar{q}_{t_{k+1}|t_k}(\cdot|x_{t_k})||p_{t_{k+1}|t_k}(\cdot|x_{t_k})) = \int_{t_k}^{t_{k+1}} \frac{\partial}{\partial t}\text{KL}(\bar{q}_{t|t_k}(\cdot|x_{t_k})||p_{t|t_k}(\cdot|x_{t_k}))\mathrm{d}t. \tag{25}$$

It suffices to provide an upper bound for the partial derivative of the KL divergence. Below, for notation brevity, we omit the conditional dependence on $x_{t_k}$ in notation and write $\bar{q}_{t|t_k}(x) =$

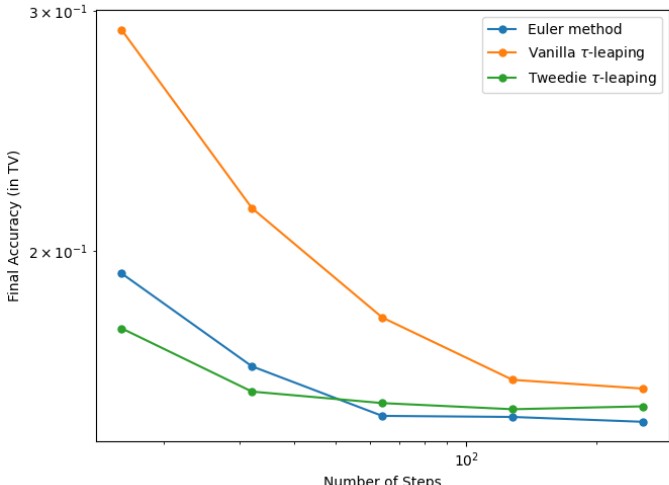

Figure 2: Estimated total variation distance between the target and sampling distribution of different sampling methods. Here $d = 3$ and $S = 8$. We use 30000 samples to estimate the TV distance.

$\breve{q}_{t|t_k}(x|x_{t_k})$ (resp. $p_{t|t_k}(x)$). We have

$$\frac{\partial}{\partial t}\mathrm{KL}(\breve{q}_{t|t_k}(\cdot|x_{t_k})||p_{t|t_k}(\cdot|x_{t_k}))$$

$$= \frac{\partial}{\partial t}\sum_{x\in\mathcal{X}}\breve{q}_{t|t_k}(x)\log\frac{\breve{q}_{t|t_k}(x)}{p_{t|t_k}(x)}$$

$$= \sum_{x\in\mathcal{X}}\left(\frac{\partial}{\partial t}\breve{q}_{t|t_k}(x)\right)\log\frac{\breve{q}_{t|t_k}(x)}{p_{t|t_k}(x)} + \sum_{x\in\mathcal{X}}\breve{q}_{t|t_k}(x)\left(\frac{\frac{\partial}{\partial t}\breve{q}_{t|t_k}(x)}{\breve{q}_{t|t_k}(x)} - \frac{\frac{\partial}{\partial t}p_{t|t_k}(x)}{p_{t|t_k}(x)}\right)$$

$$= \underbrace{\sum_{x\in\mathcal{X}}\left(\frac{\partial}{\partial t}\breve{q}_{t|t_k}(x)\right)\log\frac{\breve{q}_{t|t_k}(x)}{p_{t|t_k}(x)}}_{=:T_1} - \underbrace{\sum_{x\in\mathcal{X}}\breve{q}_{t|t_k}(x)\frac{\frac{\partial}{\partial t}p_{t|t_k}(x)}{p_{t|t_k}(x)}}_{=:T_2}.$$

Using the Kolmogorov forward equation (see (1)) and reversing $x$ and $y$, we have

$$T_1 = \sum_{x\in\mathcal{X}}\left(\sum_{y\in\mathcal{X}}\breve{R}_t(y,x)\breve{q}_{t|t_k}(y)\right)\log\frac{\breve{q}_{t|t_k}(x)}{p_{t|t_k}(x)} = \sum_{x,y\in\mathcal{X}}\breve{R}_t(x,y)\breve{q}_{t|t_k}(x)\log\frac{\breve{q}_{t|t_k}(y)}{p_{t|t_k}(y)},$$

$$T_2 = \sum_{x\in\mathcal{X}}\frac{\breve{q}_{t|t_k}(x)}{p_{t|t_k}(x)}\sum_{y\in\mathcal{X}}\hat{R}_t(y,x)p_{t|t_k}(y) = \sum_{x,y\in\mathcal{X}}\frac{\breve{q}_{t|t_k}(y)}{p_{t|t_k}(y)}\hat{R}_t(x,y)p_{t|t_k}(x).$$

In order to show the desired result, we need to relate the ratio of densities with that of the rate matrices. Recall the definition of $\breve{R}_t$ in (3). We have

$$T_1 = \sum_{x,y\in\mathcal{X}}\breve{R}_t(x,y)\breve{q}_{t|t_k}(x)\log\frac{\breve{q}_{t|t_k}(y)/\breve{q}_{t|t_k}(x)}{p_{t|t_k}(y)/p_{t|t_k}(x)} + \sum_{x\in\mathcal{X}}\breve{q}_{t|t_k}(x)\log\frac{\breve{q}_{t|t_k}(x)}{p_{t|t_k}(x)}\left(\sum_{y\in\mathcal{X}}\breve{R}_t(x,y)\right)$$

$$\overset{(i)}{=} \sum_{x,y\in\mathcal{X}}\breve{R}_t(x,y)\breve{q}_{t|t_k}(x)\log\frac{\breve{q}_{t|t_k}(y)/\breve{q}_{t|t_k}(x)}{p_{t|t_k}(y)/p_{t|t_k}(x)}$$

$$\overset{(ii)}{=} \sum_{x\in\mathcal{X}}\breve{R}_t(x,x)\breve{q}_{t|t_k}(x) + \sum_{\substack{x,y\in\mathcal{X}\\x\neq y}}\breve{R}_t(x,y)\breve{q}_{t|t_k}(x)\log\frac{\breve{R}_t(x,y)/R_{T-t|T-t_k}(y,x)}{p_{t|t_k}(y)/p_{t|t_k}(x)}$$

where $(i)$ follows because $\sum_{y\in\mathcal{X}} \breve{R}_t(x,y) = 0$, and $(ii)$ follows by Lemma 9 (where the conditioned $x_{t_k}$ is omitted for brevity) and note that $R_{T-t|T-t_k}(y,x) = \frac{q_{T-t_k|T-t}(x_{t_k}|x)}{q_{T-t_k|T-t}(x_{t_k}|y)} R_{T-t}(y,x)$. Thus,

$$
\begin{aligned}
& T_1 - T_2 \\
&= \sum_{x\in\mathcal{X}} \left( \breve{R}_t(x,x) - \hat{R}_t(x,x) \right) \bar{q}_{t|t_k}(x) \\
&\quad + \sum_{\substack{x,y\in\mathcal{X}\\x\neq y}} \bar{q}_{t|t_k}(x) \left( \breve{R}_t(x,y) \log \frac{\breve{R}_t(x,y)/R_{T-t|T-t_k}(y,x)}{p_{t|t_k}(y)/p_{t|t_k}(x)} - \frac{\bar{q}_{t|t_k}(y)/\bar{q}_{t|t_k}(x)}{p_{t|t_k}(y)/p_{t|t_k}(x)} \hat{R}_t(x,y) \right) \\
&= \sum_{x\in\mathcal{X}} \left( \breve{R}_t(x,x) - \hat{R}_t(x,x) + \sum_{\substack{y\in\mathcal{X}\\y\neq x}} \breve{R}_t(x,y) \log \frac{\breve{R}_t(x,y)}{\hat{R}_t(x,y)} \right) \bar{q}_{t|t_k}(x) \\
&\quad \underbrace{+ \sum_{\substack{x,y\in\mathcal{X}\\x\neq y}} \bar{q}_{t|t_k}(x) \left( \breve{R}_t(x,y) \log \frac{\hat{R}_t(x,y)/R_{T-t|T-t_k}(y,x)}{p_{t|t_k}(y)/p_{t|t_k}(x)} - \frac{\bar{q}_{t|t_k}(y)/\bar{q}_{t|t_k}(x)}{p_{t|t_k}(y)/p_{t|t_k}(x)} \hat{R}_t(x,y) \right)}_{=:\mathcal{R}} .
\end{aligned}
$$

Now, since $\log z \leq z - 1$ for all $z > 0$, we have

$$
\begin{aligned}
\mathcal{R} &\leq \sum_{\substack{x,y\in\mathcal{X}\\x\neq y}} \bar{q}_{t|t_k}(x) \left( \breve{R}_t(x,y) \left( \frac{\hat{R}_t(x,y)/R_{T-t|T-t_k}(y,x)}{p_{t|t_k}(y)/p_{t|t_k}(x)} - 1 \right) - \frac{\bar{q}_{t|t_k}(y)/\bar{q}_{t|t_k}(x)}{p_{t|t_k}(y)/p_{t|t_k}(x)} \hat{R}_t(x,y) \right) \\
&\overset{(iii)}{=} \sum_{\substack{x,y\in\mathcal{X}\\x\neq y}} \bar{q}_{t|t_k}(x) \left( \hat{R}_t(x,y) \left( \frac{\bar{q}_{t|t_k}(y)/\bar{q}_{t|t_k}(x)}{p_{t|t_k}(y)/p_{t|t_k}(x)} - 1 \right) - \frac{\bar{q}_{t|t_k}(y)/\bar{q}_{t|t_k}(x)}{p_{t|t_k}(y)/p_{t|t_k}(x)} \hat{R}_t(x,y) \right) \\
&\overset{(iv)}{=} \sum_{x\in\mathcal{X}} \bar{q}_{t|t_k}(x)\hat{R}_t(x,x) + \sum_{\substack{x,y\in\mathcal{X}\\x\neq y}} \bar{q}_{t|t_k}(x)\hat{R}_t(x,y) \left( \frac{\bar{q}_{t|t_k}(y)/\bar{q}_{t|t_k}(x)}{p_{t|t_k}(y)/p_{t|t_k}(x)} - \frac{\bar{q}_{t|t_k}(y)/\bar{q}_{t|t_k}(x)}{p_{t|t_k}(y)/p_{t|t_k}(x)} \right) \\
&\overset{(v)}{\leq} 0
\end{aligned}
$$

where $(iii)$ is again by Lemma 9, $(iv)$ follows because $\sum_{y\in\mathcal{X}} \hat{R}_t(x,y) = 0$, and $(v)$ follows because $\hat{R}_t(x,x) \leq 0$. Therefore,

$$
\begin{aligned}
& \mathbb{E}_{x_{t_k}\sim\bar{q}_k} \left[ \frac{\partial}{\partial t} \mathrm{KL}(\bar{q}_{t|t_k}(\cdot|x_{t_k})||p_{t|t_k}(\cdot|x_{t_k})) \right] \\
&\leq \mathbb{E}_{\substack{x_{t_k}\sim\bar{q}_k\\x_t\sim\bar{q}_{t|t_k}(\cdot|x_{t_k})}} \left[ \breve{R}_t(x_t,x_t) - \hat{R}_t(x_t,x_t) + \sum_{\substack{y\in\mathcal{X}\\y\neq x_t}} \breve{R}_t(x_t,y) \log \frac{\breve{R}_t(x_t,y)}{\hat{R}_t(x_t,y)} \right] \\
&= \mathbb{E}_{x_t\sim\bar{q}_t} \left[ \sum_{\substack{y\in\mathcal{X}\\y\neq x_t}} \hat{R}_t(x_t,y) - \breve{R}_t(x_t,y) + \breve{R}_t(x_t,y) \log \frac{\breve{R}_t(x_t,y)}{\hat{R}_t(x_t,y)} \right] .
\end{aligned}
$$

The proof is now complete by combining this with (24) and (25).

## F   Proof of Corollary 1

The proof is straight-forward by noting that

$$
\mathbb{E}_{x_t\sim\bar{q}_t} \left[ \sum_{y\neq x_t} \hat{R}_t(x_t,y) - \breve{R}_t(x_t,y) + \breve{R}_t(x_t,y) \log \frac{\breve{R}_t(x_t,y)}{\hat{R}_t(x_t,y)} \right]
$$

$$= \mathbb{E}_{x_t \sim \breve{q}_t} \left[ \sum_{y \neq x_t} R_{T-t}(y, x_t) s_{T-t}(y, x_t) - R_{T-t}(y, x_t) \frac{q_{T-t}(y)}{q_{T-t}(x_t)} \right.$$

$$\left. + R_{T-t}(y, x_t) \frac{q_{T-t}(y)}{q_{T-t}(x_t)} \log \frac{R_{T-t}(y, x_t)(q_{T-t}(y)/q_{T-t}(x_t))}{R_{T-t}(y, x_t) s_{T-t}(y, x_t)} \right]$$

$$= \mathbb{E}_{x_t \sim \breve{q}_t} \left[ \sum_{y \neq x_t} R_{T-t}(y, x_t) \left( s_{T-t}(y, x_t) - \frac{q_{T-t}(y)}{q_{T-t}(x_t)} + \frac{q_{T-t}(y)}{q_{T-t}(x_t)} \log \frac{q_{T-t}(y)/q_{T-t}(x_t)}{s_{T-t}(y, x_t)} \right) \right]$$

$$= \mathcal{L}_{SE}(T - t; s_{T-t})$$

where the last line follows from the definition of score-entropy in (4).

## G   Proof of Theorem 2

In this section, we provide the proof of Theorem 2. Before we start, the following assumption characterizes those general approximate deterministic-step-size samplers (i.e., approximation to the Kolmogorov samplers) that can sample efficiently.

**Definition 1** (Approximate Sampling Method). The sampling rate $\hat{R}_t$ is piecewise constant, i.e., constant within $t \in [t_k, t_{k+1})$. Also, given $x_{t_k} \in [S]^d$, we have $\hat{R}_t(x_{t_k}, \cdot) = \hat{R}_{t_k}(x_{t_k}, \cdot)$.

Definition 1 is especially useful for discrete diffusion models where the exact solution of the Kolmogorov equation of the sampling CTMC is computationally hard to obtain. In particular, Definition 1 is satisfied for the rate of $\tau$-leaping (see (7)). It will also be satisfied for the truncated $\tau$-leaping method later (see (30)).

### G.1   Step 1: Decomposing total error

To begin, we can employ the general result of Theorem 1 and get that

$$\mathrm{KL}(\breve{q}_{t_N} || p_{t_N})$$

$$\leq \mathrm{KL}(\breve{q}_0 || p_0) + \sum_{k=0}^{N-1} \mathbb{E}_{x_{t_k} \sim \breve{q}_k} \left[ \mathrm{KL}(\breve{q}_{t_{k+1}|t_k}(\cdot | x_{t_k}) || p_{t_{k+1}|t_k}(\cdot | x_{t_k})) \right]$$

$$\leq \mathrm{KL}(\breve{q}_0 || p_0) + \sum_{k=0}^{N-1} \int_{t_k}^{t_{k+1}} \mathbb{E}_{x_t \sim \breve{q}_t} \underbrace{\left[ \sum_{y \neq x_t} \hat{R}_t(x_t, y) - \breve{R}_t(x_t, y) + \breve{R}_t(x_t, y) \log \frac{\breve{R}_t(x_t, y)}{\hat{R}_t(x_t, y)} \right]}_{=:g_t(x_t)} \mathrm{d}t.$$

$$(26)$$

Note that $g_t$ is a Bregman divergence (generated by the negative entropy function) for each $x$, and thus $g_t(x) \geq 0$ for all $x \in [S]^d$. To see this, we fix $x_t$ and consider two vectors $p$ and $q$ such that $p_y := \breve{R}_t(x_t, y)$ and $q_y := \hat{R}_t(x_t, y)$ such that $y \neq x_t$. Also, let $\phi(p) := \sum_{y \neq x_t} p_y \log p_y$ (i.e., the negative entropy function, which is convex), and the corresponding Bregman divergence is $D_\phi(p, q) = \phi(p) - \phi(q) - \langle y - x, \phi(y) \rangle = \sum_{y \neq x_t} p_y \log p_y - q_y \log q_y - (p_y - q_y)(1 + \log q_y) = \sum_{y \neq x_t} p_y \log(p_y/q_y) - p_y + q_y$, which is exactly $g_t$. We further decompose $g_t$ into three different terms:

$$\mathrm{KL}(\breve{q}_T || p_T) \leq \mathrm{KL}(\breve{q}_0 || p_0) + \sum_{k=0}^{N-1} \int_{t_k}^{t_{k+1}} \mathbb{E}_{x_t \sim \breve{q}_t} \left[ g_t(x_t) \right] \mathrm{d}t$$

$$= \underbrace{\mathrm{KL}(\breve{q}_0 || p_0)}_{\text{initialization error}} + \underbrace{\sum_{k=0}^{N-1} (t_{k+1} - t_k) \mathbb{E}_{x_{t_k} \sim \breve{q}_{t_k}} \left[ g_{t_k}(x_{t_k}) \right]}_{\text{estimation error}} +$$

$$\underbrace{\sum_{k=0}^{N-1} \int_{t_k}^{t_{k+1}} \mathbb{E}_{\substack{x_t \sim \breve{q}_t \\ x_{t_k} \sim \breve{q}_{t_k}}} \left[ g_t(x_t) - g_t(x_{t_k}) \right] + \mathbb{E}_{x_{t_k} \sim \breve{q}_{t_k}} \left[ g_t(x_{t_k}) - g_{t_k}(x_{t_k}) \right] \mathrm{d}t}_{\text{discretization error}}. \quad (27)$$

From [19, Proposition 2] and [18, Theorem C.1], the initialization error satisfies that

$$\mathrm{KL}(\bar{q}_0 \| p_0) \lesssim (d \log S) e^{-T}.$$

Recall (3) and (5). Note that the estimation error term can be upper-bounded as

$$\sum_{k=0}^{N-1} (t_{k+1} - t_k) \mathbb{E}_{x_{t_k} \sim \bar{q}_{t_k}} \left[ g_{t_k}(x_{t_k}) \right]$$

$$= \sum_{k=0}^{N-1} (t_{k+1} - t_k) \mathbb{E}_{x_{t_k} \sim q_{T-t_k}} \left[ \sum_{y \neq x_{t_k}} \hat{R}_{t_k}(x_{t_k}, y) - \breve{R}_{t_k}(x_{t_k}, y) + \breve{R}_{t_k}(x_{t_k}, y) \log \frac{\breve{R}_{t_k}(x_{t_k}, y)}{\hat{R}_{t_k}(x_{t_k}, y)} \right]$$

$$= \sum_{k=0}^{N-1} (t_{k+1} - t_k) \mathbb{E}_{x_{t_k} \sim q_{T-t_k}} \sum_{y \neq x_{t_k}} R_{T-t_k}(y, x_{t_k}) \times$$

$$\left( s_{T-t_k}(y, x_{t_k}) - \frac{q_{T-t_k}(y)}{q_{T-t_k}(x_{t_k})} + \frac{q_{T-t_k}(y)}{q_{T-t_k}(x_{t_k})} \log \frac{q_{T-t_k}(y)/q_{T-t_k}(x_{t_k})}{s_{T-t_k}(y, x_{t_k})} \right)$$

$$= \sum_{k=0}^{N-1} (t_{k+1} - t_k) \mathcal{L}_{SE}(T - t_k; s_{T-t_k})$$

$$\leq \varepsilon_{\text{score}},$$

where the last line follows from Assumption 1. As follows, the goal is to provide an upper bound for the discretization error.

## G.2   Step 2: Identifying dominant term for discretization error

As shown in (27), the discretization error can be decomposed into two terms, one for the time-difference in the argument of $g_t$ (in expected value), and the other for the difference in $g_t$ itself. In the following lemma, we show that the former term decays faster than the other, which further implies that the latter term is the dominant error term for the discretization error.

**Lemma 1.** *For each $k = 0, \ldots, N-1$ and $t \in [t_k, t_{k+1})$, We have*

$$\mathbb{E}_{\substack{x_t \sim \bar{q}_t \\ x_{t_k} \sim \bar{q}_{t_k}}} \left[ g_t(x_t) - g_t(x_{t_k}) \right] \lesssim (t - t_k) d \cdot \mathbb{E}_{x_t \sim \bar{q}_t} \left[ g_t(x_t) \right].$$

*Proof.* See Appendix I.1. $\qquad \square$

As a result of Lemma 1, suppose that $t_{k+1} - t_k \leq \kappa$, we further have

$$\int_0^{T-\delta} \mathbb{E}_{\substack{x_t \sim \bar{q}_t \\ x_{t_k} \sim \bar{q}_{t_k}}} \left[ g_t(x_t) - g_t(x_{t_k}) \right] \leq \kappa \cdot O \left( \int_0^{T-\delta} \mathbb{E}_{x_t \sim \bar{q}_t} \left[ g_t(x_t) \right] \mathrm{d}t \right)$$

$$= \kappa \cdot O \left( \varepsilon_{\text{score}} + \int_0^{T-\delta} \mathbb{E}_{x_{t_k} \sim \bar{q}_{t_k}} \left[ g_t(x_{t_k}) - g_{t_k}(x_{t_k}) \right] \mathrm{d}t \right).$$

(28)

Here the last line follows from the decomposition of $g_t$ as in (27). Thus, this term (corresponding to the difference in $x_t$ in expectation) does not contribute to the overall rate as long as $\kappa \to 0$.

## G.3   Step 3: Bounding dominant term for discretization error

Now, we control the second term in the discretization error in (27), which is also the dominant error term as shown in Step 2. We also explicitly express its parameter dependencies. The following lemma provides a useful score bound for further analysis, which is similar to [18, Remark B.3] and [19, Lemma 2] and provided here for completeness.

**Lemma 2.** *Fix $t > 0$ and $x \neq y$ such that $\mathrm{Ham}\,(x, y) = 1$. Given the forward process in (1) with a rate given in (2), we have*

$$\frac{q_t(y)}{q_t(x)} \lesssim S \cdot \max\{1, t^{-1}\}.$$

*Proof.* See Appendix I.2. □

Now, we can upper-bound the error due to difference in $g_t$ as a difference in the likelihood ratio, as shown in the following lemma.

**Lemma 3.** *Fix $t \in [t_k, t_{k+1})$. Under Assumption 2 and Definition 1, we have*

$$\mathbb{E}_{x_{t_k} \sim \bar{q}_{t_k}} (g_t(x_{t_k}) - g_{t_k}(x_{t_k})) \lesssim (1 + \log(MS\delta^{-1})) \mathbb{E}_{x_{t_k} \sim \bar{q}_{t_k}} \sum_{y \neq x_{t_k}} \left| \breve{R}_t(x_{t_k}, y) - \breve{R}_{t_k}(x_{t_k}, y) \right|.$$

*Proof.* See Appendix I.3. □

Now, for any $x_{t_k} \in [S]^d$, the sum difference in the reverse rate can be further calculated using (3) as

$$\mathbb{E}_{x_{t_k} \sim \bar{q}_{t_k}} \sum_{y \neq x_{t_k}} \left| \breve{R}_{t_k}(x_{t_k}, y) - \breve{R}_t(x_{t_k}, y) \right|$$

$$= \mathbb{E}_{x_{t_k} \sim \bar{q}_{t_k}} \sum_{y \neq x_{t_k}} \left| \frac{q_{T-t_k}(y)}{q_{T-t_k}(x_{t_k})} R_{T-t_k}(y, x_{t_k}) - \frac{q_{T-t}(y)}{q_{T-t}(x_{t_k})} R_{T-t}(y, x_{t_k}) \right|$$

$$= \mathbb{E}_{x_{t_k} \sim \bar{q}_{t_k}} \sum_{\substack{y \neq x_{t_k} \\ \mathrm{Ham}(y, x_{t_k})=1}} \left| \frac{q_{T-t_k}(y)}{q_{T-t_k}(x_{t_k})} - \frac{q_{T-t}(y)}{q_{T-t}(x_{t_k})} \right| R_{T-t_k}(y, x_{t_k})$$

(29)

where the last line follows because $R_{T-t_k}(y, x_{t_k}) = R_{T-t}(y, x_{t_k})$ whenever $y \neq x_{t_k}$ since $\beta_t \equiv 1$.

Due to continuity of this ratio (i.e., the concrete score), one common way is to upper bound its derivative uniformly for every fixed $x$ and $y$ such that $\mathrm{Ham}(x, y) = 1$. Indeed, this is the approach taken by [18, Proposition C.2] (cf. [9, Proposition 6]). For reasons of comparison, the following upper-bound adopts the derivative-based method as in [18, Proposition C.2].

**Lemma 4** (Following the idea in [18]). *Fix $s < t$ such that $t - s$ is small. Fix $x$ and $y$ such that $\mathrm{Ham}(x, y) = 1$. Given the forward process in (1) with a rate given in (2), we have*

$$\left| \frac{q_t(y)}{q_t(x)} - \frac{q_s(y)}{q_s(x)} \right| \lesssim dS^2 \max\{1, s^{-2}\} (t - s).$$

*This further implies that*

$$\mathbb{E}_{x_{t_k} \sim \bar{q}_{t_k}} \sum_{\substack{y \neq x_{t_k} \\ \mathrm{Ham}(y, x_{t_k})=1}} \left| \frac{q_{T-t_k}(y)}{q_{T-t_k}(x_{t_k})} - \frac{q_{T-t}(y)}{q_{T-t}(x_{t_k})} \right| R_{T-t_k}(y, x_{t_k})$$

$$\lesssim d^2 S^2 \max\{1, (T - t_{k+1})^{-2}\} (t - t_k).$$

*Proof.* See Appendix I.4. □

Then, we present our novel approach below that directly provides an upper bound in expectation. This will finally result in a tighter upper bound with linear $S$ dependency.

**Lemma 5.** *Fix $s < t$ such that $t - s$ is small. Given the forward process in (1) with a rate given in (2), we have*

$$\mathbb{E}_{x_t \sim q_t} \sum_{\substack{y \neq x_t \\ \mathrm{Ham}(y, x_t)=1}} \left| \frac{q_t(y)}{q_t(x_t)} - \frac{q_s(y)}{q_s(x_t)} \right| R_t(y, x_t)$$

$$\lesssim dS \max\{1, s^{-2}\}(t - s) + d^2 S \max\{1, s^{-1}\}(t - s)$$

$$\lesssim d^2 S \max\{1, s^{-2}\}(t - s).$$

*Proof.* See Appendix I.5. □

Thus, considering the two terms in (29), we have the following bound for the expected difference in the reverse rate matrix:

$$\mathbb{E}_{x_{t_k} \sim \bar{q}_{t_k}} \sum_{y \neq x_{t_k}} \left| \bar{R}_{t_k}(x_{t_k}, y) - \bar{R}_t(x_{t_k}, y) \right| \lesssim d^2 S \max\{1, (T - t_{k+1})^{-2}\}(t - t_k).$$

Collecting the results from Steps 1–3, we would arrive at

$$\mathrm{KL}(\bar{q}_{t_N} \| p_{t_N})$$

$$\lesssim d(\log S)e^{-T} + \varepsilon_{\mathrm{score}} + d^2 S(1 + \log(MS\delta^{-1})) \sum_{k=0}^{N-1} \max\{1, (T - t_{k+1})^{-2}\}(t_{k+1} - t_k)^2.$$

Finally, to determine the overall parameter dependencies in the above summation, we can consider the particular step-size: $t_{k+1} - t_k = \kappa \min\{1, T - t_k\}$ and invoke [24, Lemma 18], which shows that

$$\sum_{k=0}^{N-1} \max\{1, (T - t_{k+1})^{-2}\}(t_{k+1} - t_k)^2 \lesssim \kappa(T + \log \delta^{-1}).$$

Also, from the last part of [17, Theorem 6], the perturbation due to early-stopping is

$$\mathrm{TV}(q_0, q_\delta) \lesssim d\delta, \quad \text{as } \delta \to 0.$$

The proof for Theorem 2 is complete.

# H    Proof of Theorem 3

The proof of Theorem 3 consists of three parts. First, we construct a non-trivial approximate discrete sampler, the truncated $\tau$-leaping algorithm, with explicit intermediate rate that allows for categorical per-step sampling. Then, we show that our truncated $\tau$-leaping is asymptotically equivalent to both the Euler method and Tweedie $\tau$-leaping in terms of the categorical sampling probabilities. Finally, we show that our proof of Theorem 2 is applicable even for such asymptotically equivalent samplers, which is further evidence of the generality of our approach.

## H.1    Step 1: Constructing an approximate sampler

To start, we propose an approximate discrete sampler that modifies the vanilla $\tau$-leaping algorithm and enables categorical sampling. We call this sampler the *truncated $\tau$-leaping* algorithm (see Appendix C). The intuition is that we only allow the first state change (a.k.a. truncated) for each dimension according to (7). This intuition is made solid by the following proposition, which shows explicitly the rate of truncated $\tau$-leaping.

**Lemma 6.** *Fix $k \in \{0, \dots, N - 1\}$ and $x_{t_k} \in [S]^d$. The truncated $\tau$-leaping algorithm corresponds to the following rate matrix: $\forall t \in [t_k, t_{k+1})$ and $\forall(x, y) : x \neq y$,*

$$\hat{R}_t^{TTL}(x, y) := \hat{R}_{t_k}(x_{t_k}, y - x + x_{t_k})\mathbb{1}\left\{\mathrm{nzind}(y - x) \in \mathrm{zeros}(x - x_{t_k})\right\}, \tag{30}$$

*where $\mathrm{nzind}(y - x)$ is the only index $i^*$ such that $x^{i^*} \neq y^{i^*}$, and $\mathrm{zeros}(v) := \{i : v^i = 0\}$ is the set of indices having zeros in a vector $v \in [S]^d$.*

*Proof.* See Appendix I.6. □

## H.2    Step 2: Establishing asymptotic equivalency

Then, we show that both the Euler method and Tweedie $\tau$-leaping are (first-order) asymptotically equivalent to truncated $\tau$-leaping. This is summarized in the following proposition.

**Lemma 7.** *Fix $t_k \in [0, T - \delta]$, $x_{t_k} \in [S]^d$, and $i \in [d]$. With some abuse of notation, write $P_{truncated}^i(a)$, $P_{tweedie}^i(a)$, and $P_{euler}^i(a)$ for the conditional probability of $x_{t_{k+1}}^i = a$ given $x_{t_k}$ for these three algorithms, respectively. Then, as $t_{k+1} - t_k \to 0$,*

$$P_{truncated}^i(a) = P_{euler}^i(a)(1 + o(1)) = P_{tweedie}^i(a)(1 + o(1)), \ \forall a \in [S].$$

*Proof.* See Appendix I.7. □

In the context of Theorem 3, note that $t_{k+1} - t_k \leq \kappa$. Thus, Lemma 7 shows that the constructed truncated $\tau$-leaping is asymptotically equivalent to both the Euler method and Tweedie $\tau$-leaping when $\kappa \to 0$ (or equivalently, when $N \to \infty$).

## H.3 Step 3: Examining the error induced by asymptotically equivalent samplers

Let $p_{t_{k+1}|t_k}$ denote the (exact) conditional probability from truncated $\tau$-leaping, and let $p'_{t_{k+1}|t_k}$ be any conditional probability such that $p'_{t_{k+1}|t_k}(\cdot|x_{t_k}) = p_{t_{k+1}|t_k}(\cdot|x_{t_k})(1 + o(1))$ for fixed $x_{t_k} \in [S]^d$, as $\kappa \to 0$. Previously, from Lemma 7, we have shown that both Euler method and Tweedie $\tau$-leaping are special cases of such $p'_{t_{k+1}|t_k}$.

A useful property for such $p'_{t_{k+1}|t_k}$ is that, for fixed $x_{t_k} \in [S]^d$, as $t_{k+1} - t_k \to 0$,

$$
\mathrm{KL}(\breve{q}_{t_{k+1}|t_k}(\cdot|x_{t_k})||p'_{t_{k+1}|t_k}(\cdot|x_{t_k}))
$$

$$
= \sum_{\tilde{x} \in [S]^d} \breve{q}_{t_{k+1}|t_k}(\tilde{x}|x_{t_k}) \log \frac{\breve{q}_{t_{k+1}|t_k}(\tilde{x}|x_{t_k})}{p'_{t_{k+1}|t_k}(\tilde{x}|x_{t_k})}
$$

$$
= \sum_{\tilde{x} \in [S]^d} \breve{q}_{t_{k+1}|t_k}(\tilde{x}|x_{t_k}) \log \frac{\breve{q}_{t_{k+1}|t_k}(\tilde{x}|x_{t_k})}{p_{t_{k+1}|t_k}(\tilde{x}|x_{t_k})} + \sum_{\tilde{x} \in [S]^d} \breve{q}_{t_{k+1}|t_k}(\tilde{x}|x_{t_k}) \log \frac{1}{1 + o(1)}
$$

$$
= \sum_{\tilde{x} \in [S]^d} \breve{q}_{t_{k+1}|t_k}(\tilde{x}|x_{t_k}) \log \frac{\breve{q}_{t_{k+1}|t_k}(\tilde{x}|x_{t_k})}{p_{t_{k+1}|t_k}(\tilde{x}|x_{t_k})} - \sum_{\tilde{x} \in [S]^d} \breve{q}_{t_{k+1}|t_k}(\tilde{x}|x_{t_k})o(1)
$$

$$
= \sum_{\tilde{x} \in [S]^d} \breve{q}_{t_{k+1}|t_k}(\tilde{x}|x_{t_k}) \log \frac{\breve{q}_{t_{k+1}|t_k}(\tilde{x}|x_{t_k})}{p_{t_{k+1}|t_k}(\tilde{x}|x_{t_k})} + o(1). \tag{31}
$$

Now we consider the decomposition in (26). We have

$$
\mathrm{KL}(\breve{q}_{t_N}||p'_{t_N})
$$

$$
\leq \mathrm{KL}(\breve{q}_0||p_0) + \sum_{k=0}^{N-1} \mathbb{E}_{x_{t_k} \sim \breve{q}_k} \left[ \mathrm{KL}(\breve{q}_{t_{k+1}|t_k}(\cdot|x_{t_k})||p'_{t_{k+1}|t_k}(\cdot|x_{t_k})) \right]
$$

$$
\lesssim \mathrm{KL}(\breve{q}_0||p_0) + \sum_{k=0}^{N-1} \mathbb{E}_{x_{t_k} \sim \breve{q}_k} \left[ \mathrm{KL}(\breve{q}_{t_{k+1}|t_k}(\cdot|x_{t_k})||p_{t_{k+1}|t_k}(\cdot|x_{t_k})) \right]
$$

where the last line follows from (31). Thus we have recovered the result in (26).

## H.4 Step 4: Examining the rate of truncated $\tau$-leaping

Now, we can verify that the rate matrix in (30) satisfies Definition 1, just as vanilla $\tau$-leaping. Thus, the rate of Theorem 2 still holds if we substitute $\tau$-leaping with truncated $\tau$-leaping. The proof of Theorem 3 is now complete.

# I   Proofs of Auxiliary Lemmas

## I.1 Proof of Lemma 1

With the forward process in (1), we have

$$
\mathbb{E}_{\substack{x_t \sim \breve{q}_t \\ x_{t_k} \sim \breve{q}_{t_k}}} \left[ g_t(x_t) - g_t(x_{t_k}) \right]
$$

$$
= \mathbb{E}_{x_t \sim \breve{q}_t} \left[ g_t(x_t) - \sum_{x_{t_k} \in [S]^d} q_{T-t_k|T-t}(x_{t_k}|x_t)g_t(x_{t_k}) \right]
$$

$$
= \mathbb{E}_{x_t \sim \breve{q}_t} \left[ g_t(x_t) - \sum_{x_{t_k} \in [S]^d} \left( \mathbb{1}\{x_{t_k} = x_t\} + R_t(x_t, x_{t_k})(t - t_k) \right) g_t(x_{t_k}) \right] + o(t - t_k)
$$

$$
= (t - t_k)\mathbb{E}_{x_t \sim \breve{q}_t} \left[ -\sum_{x_{t_k} \in [S]^d} R_t(x_t, x_{t_k})g_t(x_{t_k}) \right] + o(t - t_k)
$$

$$
\overset{(i)}{\leq} (t - t_k)\mathbb{E}_{x_t \sim \breve{q}_t} \left[ (-R_t(x_t, x_t))g_t(x_t) \right] + o(t - t_k)
$$

$$= (t - t_k) \frac{S-1}{S} d \cdot \mathbb{E}_{x_t \sim \bar{q}_t} \left[ g_t(x_t) \right] + o(t - t_k)$$

where $(i)$ follows because $g_t(x) \geq 0$ and $R_t(x, y) \geq 0$ if $x \neq y$. Also, for the last line, note that $R_t(x, x) = -\sum_{y \neq x} R_t(x, y) = -\frac{S-1}{S} d$ when $\beta_t \equiv 1$. The proof is now complete.

## I.2 Proof of Lemma 2

Let $j$ be the only index such that $x^j \neq y^j$. First, we note that

$$
\begin{aligned}
\frac{q_t(y)}{q_t(x)} &= \frac{1}{q_t(x)} \sum_{x_0 \in [S]^d} q_0(x_0) q_{t|0}(y|x_0) \\
&\stackrel{(i)}{=} \frac{1}{q_t(x)} \sum_{x_0 \in [S]^d} q_0(x_0) \prod_{i \in [d]} q_{t|0}^i(y^i|x_0^i) \\
&= \frac{1}{q_t(x)} \sum_{x_0 \in [S]^d} q_0(x_0) \left( \prod_{i \in [d]} q_{t|0}^i(x^i|x_0^i) \right) \left( \frac{q_{t|0}^j(y^j|x_0^j)}{q_{t|0}^j(x^j|x_0^j)} \right) \\
&\stackrel{(ii)}{=} \sum_{x_0 \in [S]^d} \frac{q_0(x_0) q_{t|0}(x|x_0)}{q_t(x)} \left( \frac{q_{t|0}^j(y^j|x_0^j)}{q_{t|0}^j(x^j|x_0^j)} \right) \\
&= \mathbb{E}_{x_0 \sim q_{0|t}(\cdot|x)} \frac{q_{t|0}^j(y^j|x_0^j)}{q_{t|0}^j(x^j|x_0^j)}
\end{aligned}
\tag{32}
$$

where both $(i)$ and $(ii)$ follow because with the chosen $R_t$ in (2) each dimension propagates independently in the forward process (cf. [9, Prop. 3]). Note that the reverse process does not propagate independently.

To obtain an analytical solution for the conditional probability, we can solve the Kolmogorov forward equation for the $i$-th dimension ($\forall i \in [d]$) (cf. [19, Proposition 1]):

$$\frac{\mathrm{d}}{\mathrm{d}t} q_{t|0}^i(z|a) = \sum_{\tilde{z} \in [S]} q_{t|0}^i(\tilde{z}|a) R_t^{\mathrm{tok}}(\tilde{z}, z),$$

whose solution is

$$
\begin{aligned}
q_{t|0}^i(z|a) &= \left[ \exp \left( \int_0^t R_s^{\mathrm{tok}} \mathrm{d}s \right) \right] (a, z) = \left[ \exp \left( t R_{\mathrm{base}} \right) \right] (a, z) = \left[ P \exp \left( t \Lambda \right) P^{-1} \right] (a, z) \\
&= \begin{cases} S^{-1}(1 - e^{-t}) & \text{if } z \neq a \\ S^{-1}(1 + (S-1)e^{-t}) & \text{if } z = a \end{cases}
\end{aligned}
\tag{33}
$$

where we recall that $R_t^{\mathrm{tok}} = R_{\mathrm{base}}$ when $\beta_t \equiv 1$ and we denote the eigendecomposition of $R_{\mathrm{base}} = S^{-1} \mathbf{1}_S \mathbf{1}_S^{\mathsf{T}} - I_S$ as $R_{\mathrm{base}} = P \Lambda P^{-1}$. Thus, plugging back into (32), we have

$$\frac{q_{t|0}^j(y^j|x_0^j)}{q_{t|0}^j(x^j|x_0^j)} = \begin{cases} 1 & \text{if } x^j \neq x_0^j \text{ and } y^j \neq x_0^j \\ \frac{1-e^{-t}}{1+(S-1)e^{-t}} & \text{if } x^j = x_0^j \text{ but } y^j \neq x_0^j \\ \frac{1+(S-1)e^{-t}}{1-e^{-t}} & \text{if } x^j \neq x_0^j \text{ but } y^j = x_0^j \end{cases}.
\tag{34}$$

Among the three cases above, since $e^{-t} \geq 0$, the second case satisfies that $\frac{1-e^{-t}}{1+(S-1)e^{-t}} \leq 1$. Also, the third case satisfies that

$$
\begin{aligned}
\frac{1 + (S-1)e^{-t}}{1 - e^{-t}} &= 1 + S \cdot \frac{1}{e^t - 1} \leq \begin{cases} S+1 & \text{if } e^t \geq 2 \\ \frac{S}{t} & \text{otherwise} \end{cases} \\
&\lesssim S \cdot \max\{1, t^{-1}\}.
\end{aligned}
$$

Therefore, the bound is as desired.

## I.3 Proof of Lemma 3

By definition of $g_t$, we have

$$\mathbb{E}_{x_{t_k}\sim\bar{q}_{t_k}}\left(g_t(x_{t_k}) - g_{t_k}(x_{t_k})\right)$$

$$= \mathbb{E}_{x_{t_k}\sim\bar{q}_{t_k}} \sum_{y\neq x_{t_k}} \left(\hat{R}_t(x_{t_k},y) - \hat{R}_{t_k}(x_{t_k},y)\right) - \left(\breve{R}_t(x_{t_k},y) - \breve{R}_{t_k}(x_{t_k},y)\right)$$

$$+ \left(\breve{R}_t(x_{t_k},y)\log\frac{\breve{R}_t(x_{t_k},y)}{\hat{R}_t(x_{t_k},y)} - \breve{R}_{t_k}(x_{t_k},y)\log\frac{\breve{R}_{t_k}(x_{t_k},y)}{\hat{R}_{t_k}(x_{t_k},y)}\right)$$

$$\overset{(i)}{=} \mathbb{E}_{x_{t_k}\sim\bar{q}_{t_k}} \sum_{y\neq x_{t_k}} \left(\breve{R}_t(x_{t_k},y)\log\frac{\breve{R}_t(x_{t_k},y)}{\hat{R}_{t_k}(x_{t_k},y)} - \breve{R}_{t_k}(x_{t_k},y)\log\frac{\breve{R}_{t_k}(x_{t_k},y)}{\hat{R}_{t_k}(x_{t_k},y)}\right)$$

$$- \left(\breve{R}_t(x_{t_k},y) - \breve{R}_{t_k}(x_{t_k},y)\right)$$

$$\leq \mathbb{E}_{x_{t_k}\sim\bar{q}_{t_k}} \sum_{y\neq x_{t_k}} \left|\breve{R}_t(x_{t_k},y) - \breve{R}_{t_k}(x_{t_k},y)\right|\left(1 + \left|\log\hat{R}_{t_k}(x_{t_k},y)\right|\right)$$

$$+ \left|\breve{R}_t(x_{t_k},y)\log\breve{R}_t(x_{t_k},y) - \breve{R}_{t_k}(x_{t_k},y)\log\breve{R}_{t_k}(x_{t_k},y)\right|$$

$$\overset{(ii)}{\lesssim} (1 + \log(MS\delta^{-1})) \cdot \mathbb{E}_{x_{t_k}\sim\bar{q}_{t_k}} \sum_{y\neq x_{t_k}} \left|\breve{R}_t(x_{t_k},y) - \breve{R}_{t_k}(x_{t_k},y)\right|$$

where $(i)$ follows by Definition 1. We explain $(ii)$ as follows. First, from (5), if $x \neq y$,

$$\hat{R}_{t_k}(x,y) = R_{T-t_k}(y,x)s_{T-t_k}(y,x) \in [M^{-1}S^{-1}, MS^{-1}]$$

under Assumption 2, which further implies that $\left|\log\hat{R}_{t_k}(x,y)\right| \leq \log(MS)$. Also,

$$\left|\breve{R}_t(x_{t_k},y)\log\breve{R}_t(x_{t_k},y) - \breve{R}_{t_k}(x_{t_k},y)\log\breve{R}_{t_k}(x_{t_k},y)\right|$$

$$\overset{(iii)}{\leq} \left(1 + \left|\log\breve{R}^*\right|\right)\left|\breve{R}_t(x_{t_k},y) - \breve{R}_{t_k}(x_{t_k},y)\right|$$

$$\overset{(iv)}{\lesssim} \left(1 + \log(S\delta^{-1})\right)\left|\breve{R}_t(x_{t_k},y) - \breve{R}_{t_k}(x_{t_k},y)\right|$$

where $(iii)$ follows from the intermediate-value theorem for $f(z) = z\log z$ and $\breve{R}^*$ is a number between $\breve{R}_{t_k}(x_{t_k},y)$ and $\breve{R}_t(x_{t_k},y)$, and $(iv)$ follows because, by Lemma 2, $\breve{R}_t(x,y) = R_{T-t}(y,x)\frac{q_{T-t}(y)}{q_{T-t}(x)} \lesssim \max\{1,(T-t)^{-1}\} \leq \delta^{-1}$ for all $t > 0$ if $x \neq y$. Meanwhile, by symmetry, $\breve{R}_t(x,y) \gtrsim \frac{1}{S^2\max\{1,(T-t)^{-1}\}} \geq \frac{\delta}{S^2}$. The proof is now complete.

## I.4 Proof of Lemma 4

The idea comes from [18, Proposition C.2]. When $t - s$ is small, we note that the derivative of the likelihood ratio w.r.t. $t$ is equal to

$$\left|\frac{\partial}{\partial t}\left(\frac{q_t(y)}{q_t(x)}\right)\right| = \left|\frac{\frac{\partial}{\partial t}q_t(y)}{q_t(x)} - \frac{q_t(y)\frac{\partial}{\partial t}q_t(x)}{q_t(x)^2}\right| \leq \frac{\left|\frac{\partial}{\partial t}q_t(y)\right|}{q_t(x)} + \frac{q_t(y)\left|\frac{\partial}{\partial t}q_t(x)\right|}{q_t(x)^2}.$$

Now, by Kolmogorov forward equation,

$$\left|\frac{\frac{\partial}{\partial t}q_t(y)}{q_t(x)}\right| = \left|\sum_{y'\in[S]^d}\frac{q_t(y')}{q_t(x)}R_t(y',y)\right|$$

$$= \frac{1}{S}\cdot\frac{q_t(y)}{q_t(x)}\sum_{\substack{y'\in[S]^d \\ \mathrm{Ham}(y,y')=1}}\frac{q_t(y')}{q_t(y)} + d\frac{S-1}{S}\frac{q_t(y)}{q_t(x)}$$

$$\overset{(i)}{\lesssim} dS^2 \max\{1, t^{-1}\}^2 + dS \max\{1, t^{-1}\}$$
$$\lesssim dS^2 \max\{1, t^{-1}\}^2$$

where $(i)$ follows from Lemma 2 and note that the summation has $d(S-1)$ terms in total. With a similar argument, we have

$$\left| \frac{\frac{\partial}{\partial t} q_t(x)}{q_t(x)} \right| = \frac{1}{S} \cdot \sum_{\substack{x' \in [S]^d \\ \mathrm{Ham}(x,x')=1}} \frac{q_t(x')}{q_t(x)} + d\frac{S-1}{S} \lesssim dS \max\{1, t^{-1}\}.$$

Therefore,

$$\left| \frac{\partial}{\partial t} \left( \frac{q_t(y)}{q_t(x)} \right) \right| \lesssim dS^2 \max\{1, t^{-2}\},$$

and thus

$$\left| \frac{q_t(y)}{q_t(x)} - \frac{q_s(y)}{q_s(x)} \right| \lesssim \left| \frac{\partial}{\partial t} \left( \frac{q_t(y)}{q_t(x)} \right) \right| (t-s) \lesssim dS^2 \max\{1, t^{-2}\}(t-s),$$

as claimed. Especially note the quadratic dependency on $S$.

## I.5 Proof of Lemma 5

First, fix $x_t$ and $y$ and let $i$ be the index such that $x_t^i \neq y^i$. From (32), we have,

$$\left| \frac{q_t(y)}{q_t(x_t)} - \frac{q_s(y)}{q_s(x_t)} \right| = \left| \mathbb{E}_{x_0 \sim q_{0|t}(\cdot|x_t)} \left[ \frac{q_{t|0}^i(y^i|x_0^i)}{q_{t|0}^i(x_t^i|x_0^i)} \right] - \mathbb{E}_{\tilde{x}_0 \sim q_{0|s}(\cdot|x_t)} \left[ \frac{q_{s|0}^i(y^i|\tilde{x}_0^i)}{q_{s|0}^i(x_t^i|\tilde{x}_0^i)} \right] \right|$$

$$\leq \mathbb{E}_{x_0 \sim q_{0|t}(\cdot|x_t)} \left| \frac{q_{t|0}^i(y^i|x_0^i)}{q_{t|0}^i(x_t^i|x_0^i)} - \frac{q_{s|0}^i(y^i|x_0^i)}{q_{s|0}^i(x_t^i|x_0^i)} \right|$$

$$+ \left| \mathbb{E}_{\substack{x_0 \sim q_{0|t}(\cdot|x_t) \\ \tilde{x}_0 \sim q_{0|s}(\cdot|x_t)}} \left[ \frac{q_{s|0}^i(y^i|x_0^i)}{q_{s|0}^i(x_t^i|x_0^i)} - \frac{q_{s|0}^i(y^i|\tilde{x}_0^i)}{q_{s|0}^i(x_t^i|\tilde{x}_0^i)} \right] \right|. \qquad (35)$$

For the first term in (35), we note the expression of likelihood ratio in (34) and thus, for any fixed $x_0$, $x_t$, and $y$,

$$\left| \frac{q_{t|0}^i(y^i|x_0^i)}{q_{t|0}^i(x_t^i|x_0^i)} - \frac{q_{s|0}^i(y^i|x_0^i)}{q_{s|0}^i(x_t^i|x_0^i)} \right| = \begin{cases} 0 & \text{if } x^i \neq x_0^i \text{ and } y^i \neq x_0^i \\ \frac{1-e^{-t}}{1+(S-1)e^{-t}} - \frac{1-e^{-s}}{1+(S-1)e^{-s}} & \text{if } x^i = x_0^i \text{ but } y^i \neq x_0^i \\ \frac{1+(S-1)e^{-t}}{1-e^{-t}} - \frac{1+(S-1)e^{-s}}{1-e^{-s}} & \text{if } x^i \neq x_0^i \text{ but } y^i = x_0^i \end{cases}.$$

Now, since

$$\left| \frac{\partial}{\partial t} \left( \frac{1-e^{-t}}{1+(S-1)e^{-t}} \right) \right| = \frac{Se^t}{(S+e^t-1)^2} \lesssim 1$$

$$\left| \frac{\partial}{\partial t} \left( \frac{1+(S-1)e^{-t}}{1-e^{-t}} \right) \right| = \frac{Se^t}{(e^t-1)^2} \lesssim \frac{S}{\min\{1,t\}^2},$$

we have

$$\left| \frac{q_{t|0}^i(y^i|x_0^i)}{q_{t|0}^i(x_t^i|x_0^i)} - \frac{q_{s|0}^i(y^i|x_0^i)}{q_{s|0}^i(x_t^i|x_0^i)} \right| \lesssim \frac{S}{\min\{1,t\}^2}(t-s).$$

Note that this term does not depend on $d$. Thus,

$$\mathbb{E}_{x_t \sim q_t} \sum_{\substack{y \neq x_t \\ \mathrm{Ham}(y,x_t)=1}} \mathbb{E}_{x_0 \sim q_{0|t}(\cdot|x_t)} \left| \frac{q_{t|0}^i(y^i|x_0^i)}{q_{t|0}^i(x_t^i|x_0^i)} - \frac{q_{s|0}^i(y^i|x_0^i)}{q_{s|0}^i(x_t^i|x_0^i)} \right| R_t(y, x_t)$$

$$\lesssim dS \max\{1, t^{-2}\}(t-s).$$

Now we turn to the second term in (35). Write $f(z) := \frac{q^i_{s|0}(y^i|z)}{q^i_{s|0}(x^i_t|z)}$ for brevity (recall that $x_t$ and $y$ are fixed and thus omitted in this expression). Note that from (34), an upper bound on $f(z)$ is

$$\sup_{y^i,x^i_t,z\in[S]} f(z) = \sup_{y^i,x^i_t,z\in[S]} \frac{q^i_{s|0}(y^i|z)}{q^i_{s|0}(x^i_t|z)} \lesssim S \cdot \max\{1, s^{-1}\}.$$

Thus, the second term in (35) can be upper-bounded (for each $y^i$) as

$$\left| \mathbb{E}_{\substack{x_0\sim q_{0|t}(\cdot|x_t) \\ \tilde{x}_0\sim q_{0|s}(\cdot|x_t)}} \left[ f(x^i_0) - f(\tilde{x}^i_0) \right] \right| = \left| \sum_{x_0\in[S]^d} f(x^i_0)(q_{0|t}(x_0|x_t) - q_{0|s}(x_0|x_t)) \right|$$

$$\lesssim S \max\{1, s^{-1}\} \sum_{x_0\in[S]^d} \left| q_{0|t}(x_0|x_t) - q_{0|s}(x_0|x_t) \right|. \qquad (36)$$

Using Bayes' rule, we have

$$\sum_{x_0\in[S]^d} \left| q_{0|t}(x_0|x_t) - q_{0|s}(x_0|x_t) \right| = \sum_{x_0\in[S]^d} q_0(x_0) \left| \frac{q_{t|0}(x_t|x_0)}{q_t(x_t)} - \frac{q_{s|0}(x_t|x_0)}{q_s(x_t)} \right|$$

$$\leq \frac{1}{q_t(x_t)\cdot q_s(x_t)} \sum_{x_0,y_0\in[S]^d} q_0(x_0)q_0(y_0) \left| q_{t|0}(x_t|x_0)q_{s|0}(x_t|y_0) - q_{s|0}(x_t|x_0)q_{t|0}(x_t|y_0) \right|$$

$$\leq \frac{1}{q_t(x_t)\cdot q_s(x_t)} \mathbb{E}_{x_0,y_0\sim q_0} \left[ \left| q_{t|0}(x_t|x_0) - q_{s|0}(x_t|x_0) \right| q_{s|0}(x_t|y_0) \right.$$

$$\left. + \left| q_{s|0}(x_t|y_0) - q_{t|0}(x_t|y_0) \right| q_{s|0}(x_t|x_0) \right]$$

$$= \frac{1}{q_t(x_t)} \mathbb{E}_{x_0\sim q_0} \left| q_{t|0}(x_t|x_0) - q_{s|0}(x_t|x_0) \right| + \frac{1}{q_t(x_t)} \mathbb{E}_{y_0\sim q_0} \left| q_{s|0}(x_t|y_0) - q_{t|0}(x_t|y_0) \right|$$

$$= \frac{2}{q_t(x_t)} \mathbb{E}_{x_0\sim q_0} \left| q_{t|0}(x_t|x_0) - q_{s|0}(x_t|x_0) \right|. \qquad (37)$$

Now, this term (without the constant factor 2) can be upper-bounded as

$$\frac{1}{q_t(x_t)} \mathbb{E}_{x_0\sim q_0} \left| q_{t|0}(x_t|x_0) - q_{s|0}(x_t|x_0) \right|$$

$$\lesssim \frac{1}{q_t(x_t)}(t-s)\mathbb{E}_{x_0\sim q_0} \left| \frac{\partial}{\partial t} q_{t|0}(x_t|x_0) \right|$$

$$\overset{(i)}{=} \frac{1}{q_t(x_t)}(t-s)\mathbb{E}_{x_0\sim q_0} \left| \sum_{\tilde{x}_t\in[S]^d} q_{t|0}(\tilde{x}_t|x_0)R_t(\tilde{x}_t, x_t) \right|$$

$$\leq \frac{1}{q_t(x_t)}(t-s)\mathbb{E}_{x_0\sim q_0} \sum_{\tilde{x}_t\in[S]^d} q_{t|0}(\tilde{x}_t|x_0) \left| R_t(\tilde{x}_t, x_t) \right|$$

$$= (t-s) \sum_{\tilde{x}_t\in[S]^d} \frac{q_t(\tilde{x}_t)}{q_t(x_t)} \left| R_t(\tilde{x}_t, x_t) \right|, \qquad (38)$$

where $(i)$ follows from Kolmogorov forward equation. Thus, combining these intermediate results, we have

$$\mathbb{E}_{x_t\sim q_t} \sum_{\substack{y\neq x_t \\ \mathrm{Ham}(y,x_t)=1}} \left| \mathbb{E}_{\substack{x_0\sim q_{0|t}(\cdot|x_t) \\ \tilde{x}_0\sim q_{0|s}(\cdot|x_t)}} \left[ f(x^i_0) - f(\tilde{x}^i_0) \right] \right| R_t(y, x_t)$$

$$\overset{(ii)}{\lesssim} dS \max\{1, s^{-1}\} \cdot \mathbb{E}_{x_t\sim q_t} \left[ \frac{1}{q_t(x_t)} \mathbb{E}_{x_0\sim q_0} \left| q_{t|0}(x_t|x_0) - q_{s|0}(x_t|x_0) \right| \right]$$

$$\overset{(iii)}{\lesssim} (t-s)dS\max\{1,s^{-1}\}\mathbb{E}_{x_t\sim q_t}\left[\sum_{\tilde{x}_t\in[S]^d}\frac{q_t(\tilde{x}_t)}{q_t(x_t)}|R_t(\tilde{x}_t,x_t)|\right]$$

$$= (t-s)dS\max\{1,s^{-1}\}\mathbb{E}_{\tilde{x}_t\sim q_t}\sum_{x_t\in[S]^d}|R_t(\tilde{x}_t,x_t)|$$

$$\overset{(iv)}{\asymp} (t-s)d^2S\max\{1,s^{-1}\},$$

where $(ii)$ follows from (36) and (37), $(iii)$ follows from (38), and $(iv)$ follows because $-R_t(x,x) = \sum_{y\neq x}R_t(x,y) = \frac{S-1}{S}d$ for all $x,y\in[S]^d$. The proof is now complete.

### I.6  Proof of Lemma 6

Note that $\hat{R}_t^{\text{TTL}}(x,y)\equiv 0$ whenever $\text{Ham}(x,y)\geq 2$ by definition of $\hat{R}_{t_k}$. Also recall the definition of the token-wise rate $\hat{R}_k^i$ in (9). Note that $\hat{R}_k^i$ is a valid rate matrix since $\sum_{a\in[S]}\hat{R}_k^i(z,a) = 0$ for all $z\in[S]$ and $\hat{R}_k^i(z,a)\geq 0$ if $z\neq a$.

Fix $x$ and $y$ such that $\text{Ham}(x,y) = 1$. Let $i^*\in[d]$ be the (only) index such that $y^{i^*}\neq x^{i^*}$. We now divide into the following three cases:

1. Case 1: $x = x_{t_k}$. Then, $\text{zeros}(x - x_{t_k}) = [S]$, and $\hat{R}_t^{\text{TTL}}(x,y) = \hat{R}_{t_k}(x_{t_k},y) = \hat{R}_k^{i^*}(x_{t_k}^{i^*},y^{i^*})$.

2. Case 2: $x\neq x_{t_k}$, but $x^{i^*} = x_{t_k}^{i^*}$. Thus, $i^*\in\text{zeros}(x - x_{t_k})$, and $\hat{R}_t^{\text{TTL}}(x,y) = \hat{R}_{t_k}(x_{t_k},x_{t_k}^{-i^*}\oplus_{i^*}y^{i^*}) = \hat{R}_k^{i^*}(x_{t_k}^{i^*},y^{i^*})$.

3. Case 3: $x^{i^*}\neq x_{t_k}^{i^*}$ (and also $x\neq x_{t_k}$). Then, $i^*\notin\text{zeros}(x - x_{t_k})$, and $\hat{R}_t^{\text{TTL}}(x,y) = 0$.

Thus, the overall CTMC is equivalent to $S$ CTMC's, one for each dimension. The transition rate matrix on the $i$-th dimension is $\hat{R}_k^i(z,a)\mathbb{1}\{z = x_{t_k}^i\}$. Notably, at most one state transition can happen during $t\in[t_k,t_{k+1})$ (where the CTMC stops after its first transition). We can also calculate the corresponding state transition probability by solving the Kolmogorov forward equation, which is equal to (for example, see [52, Chap. 2])

$$P_{t_{k+1}|t_k}^i\left\{x_{t_{k+1}}^i = a|x_{t_k}\right\} = \begin{cases} \exp\left(\hat{R}_k^i(x_{t_k}^i,x_{t_k}^i)(t_{k+1}-t_k)\right) & \text{if } a = x_{t_k}^i \\ \frac{\hat{R}_k^i(x_{t_k}^i,a)}{-\hat{R}_k^i(x_{t_k}^i,x_{t_k}^i)}\left(1 - \exp\left(\hat{R}_k^i(x_{t_k}^i,x_{t_k}^i)(t_{k+1}-t_k)\right)\right) & \text{if } a\neq x_{t_k}^i \end{cases}.$$

This is the same as the transition in (23). The proof is now complete.

### I.7  Proof of Lemma 7

Recall the conditional probabilities from (23), (8), and (10). Also there we have defined $\hat{R}_k^i$ such that

$$\hat{R}_k^i(x_{t_k}^i,a) = \hat{R}_{t_k}(x_{t_k},x_{t_k}^{-i}\oplus_i a),\quad \forall a\neq x_{t_k}^i.$$

As follows, we only consider all $a\in[S]$ such that $a\neq x_{t_k}^i$, since all probability vectors need to sum up as 1.

We first focus on the Euler method. From (23),

$$P_{\text{truncated}}^i(a) = \frac{\hat{R}_k^i(x_{t_k}^i,a)}{-\hat{R}_k^i(x_{t_k}^i,x_{t_k}^i)}\left(1 - \exp\left(\hat{R}_k^i(x_{t_k}^i,x_{t_k}^i)\cdot(t_{k+1}-t_k)\right)\right)$$

$$= \frac{\hat{R}_k^i(x_{t_k}^i,a)}{-\hat{R}_k^i(x_{t_k}^i,x_{t_k}^i)}(-\hat{R}_k^i(x_{t_k}^i,x_{t_k}^i)\cdot((t_{k+1}-t_k)+o(t_{k+1}-t_k)))$$

$$= \hat{R}_k^i(x_{t_k}^i,a)(t_{k+1}-t_k)(1+o(1))$$

$$= P_{\text{euler}}^i(a)(1+o(1)).$$

Also, for Tweedie $\tau$-leaping, from (10), we have

$$P_{\text{tweedie}}^i(a)$$

$$= \left( \left[ e^{-(\bar{\beta}_{T-t_k} - \bar{\beta}_{T-t_{k+1}})R_{\text{base}}} \right]^{a:} s_{T-t_k}(x_{t_k}^{-i} \oplus_i \cdot, x_{t_k}) \right) \left[ e^{(\bar{\beta}_{T-t_k} - \bar{\beta}_{T-t_{k+1}})R_{\text{base}}} \right]^{a,x_{t_k}^i}$$

$$= \left[ \delta_a - (\bar{\beta}_{T-t_k} - \bar{\beta}_{T-t_{k+1}})R_{\text{base}}(a, \cdot) \right] s_{T-t_k}(x_{t_k}^{-i} \oplus_i \cdot, x_{t_k})(\bar{\beta}_{T-t_k} - \bar{\beta}_{T-t_{k+1}})R_{\text{base}}(a, x_{t_k}^i)(1 + o(1))$$

$$= s_{T-t_k}(x_{t_k}^{-i} \oplus_i a, x_{t_k})(\bar{\beta}_{T-t_k} - \bar{\beta}_{T-t_{k+1}})R_{\text{base}}(a, x_{t_k}^i)(1 + o(1))$$

$$= s_{T-t_k}(x_{t_k}^{-i} \oplus_i a, x_{t_k})(t_{k+1} - t_k)\beta_{T-t_k}R_{\text{base}}(a, x_{t_k}^i)(1 + o(1))$$

$$= s_{T-t_k}(x_{t_k}^{-i} \oplus_i a, x_{t_k})R_{T-t_k}(x_{t_k}^{-i} \oplus_i a, x_{t_k})(t_{k+1} - t_k)(1 + o(1))$$

$$= \hat{R}_k^i(x_{t_k}^i, a)(t_{k+1} - t_k)(1 + o(1))$$

$$= P_{\text{euler}}^i(a)(1 + o(1)),$$

where $\delta_a$ is such that $\text{Ham}(\delta_a, 0) = 1$ and $[\delta_a]^a = 1$. The proof is now complete.

## I.8  Lemma 8 and its proof

**Lemma 8.** *Fix $a \in [S]^d$. We have*

$$\lim_{s \downarrow t_k} \text{KL}(\bar{q}_{s|t_k}(\cdot|a) || p_{s|t_k}(\cdot|a)) = 0.$$

*Proof of Lemma 8.* First, we have

$$\lim_{s \downarrow t_k} \text{KL}(\bar{q}_{s|t_k}(\cdot|a) || p_{s|t_k}(\cdot|a))$$

$$= \lim_{s \downarrow t_k} \sum_{x \in \mathcal{X}} \bar{q}_{s|t_k}(x|a) \log \frac{\bar{q}_{s|t_k}(x|a)}{p_{s|t_k}(x|a)}$$

$$\stackrel{(*)}{=} \sum_{x \in \mathcal{X}} \lim_{s \downarrow t_k} \left( \bar{q}_{s|t_k}(x|a) \log \frac{\bar{q}_{s|t_k}(x|a)}{p_{s|t_k}(x|a)} \right)$$

$$= \sum_{x \in \mathcal{X}} \left( \lim_{s \downarrow t_k} \bar{q}_{s|t_k}(x|a) \right) \left( \lim_{s \downarrow t_k} \log \frac{\bar{q}_{s|t_k}(x|a)}{p_{s|t_k}(x|a)} \right).$$

Here $(*)$ follows because, different from the case where $\mathcal{X} = \mathbb{R}^d$, we can safely interchange the limit and summation because $\mathcal{X}$ has finite cardinality. Now, by definition of the CTMC process (see (1)),

$$\lim_{s \downarrow t_k} \bar{q}_{s|t_k}(x|a) = \lim_{s \downarrow t_k} p_{s|t_k}(x|a) = \mathbb{1}\{x = a\},$$

which implies the desired result. $\square$

## I.9  Lemma 9 and its proof

**Lemma 9.** *Fix $s < t$ (thus $T - s > T - t$) and $a \in [S]^d$. We have*

$$\bar{R}_t(x, y) = \frac{\bar{q}_{t|s}(y|a)}{\bar{q}_{t|s}(x|a)} R_{T-t|T-s}(y, x|a), \quad \forall x \neq y.$$

*Here $R_{T-t|T-s}(\cdot, \cdot|a)$ is defined as the rate matrix for the forward CTMC at time $T - t$ conditioned on the future observation at time $T - s$ being $a$. Indeed, we further have*

$$R_{T-t|T-s}(y, x|a) = \frac{q_{T-s|T-t}(a|x)}{q_{T-s|T-t}(a|y)} R_{T-t}(y, x) \in [0, \infty).$$

*Proof of Lemma 9.* Fix $x \neq y$. First, by Bayes' rule, $\forall \tilde{t} \in (s, t)$ (and thus $T - s > T - \tilde{t} > T - t$),

$$\bar{q}_{t|\tilde{t},s}(y|x, a) = q_{T-t|T-\tilde{t},T-s}(y|x, a) = q_{T-\tilde{t}|T-t,T-s}(x|y, a) \cdot \frac{q_{T-t|T-s}(y|a)}{q_{T-\tilde{t}|T-s}(x|a)}.$$

For the left-hand side, by the Markov property of the reverse process, we have that

$$\lim_{\tilde{t} \to t} \frac{\partial}{\partial t} \bar{q}_{t|\tilde{t},s}(y|x, a) = \lim_{\tilde{t} \to t} \frac{\partial}{\partial t} \bar{q}_{t|\tilde{t}}(y|x) = \bar{R}_t(x, y).$$

For the right-hand side, we note that

$$\lim_{\tilde{t} \to t} \frac{\partial}{\partial t} \left( q_{T-\tilde{t}|T-t,T-s}(x|y,a) \cdot \frac{q_{T-t|T-s}(y|a)}{q_{T-\tilde{t}|T-s}(x|a)} \right)$$

$$= \lim_{\tilde{t} \to t} \left( \frac{\partial}{\partial t} q_{T-\tilde{t}|T-t,T-s}(x|y,a) \right) \frac{q_{T-t|T-s}(y|a)}{q_{T-\tilde{t}|T-s}(x|a)} + \lim_{\tilde{t} \to t} q_{T-\tilde{t}|T-t,T-s}(x|y,a) \frac{\frac{\partial}{\partial t} q_{T-t|T-s}(y|a)}{q_{T-\tilde{t}|T-s}(x|a)}$$

$$\overset{(i)}{=} \lim_{\tilde{t} \to t} \left( \frac{\partial}{\partial t} q_{T-\tilde{t}|T-t,T-s}(x|y,a) \right) \frac{q_{T-t|T-s}(y|a)}{q_{T-\tilde{t}|T-s}(x|a)}$$

$$= \left( \lim_{\tilde{t} \to t} \frac{\partial}{\partial t} q_{T-\tilde{t}|T-t,T-s}(x|y,a) \right) \frac{q_{T-t|T-s}(y|a)}{q_{T-t|T-s}(x|a)}$$

where $(i)$ follows because $\lim_{\tilde{t} \to t} q_{T-\tilde{t}|T-t,T-s}(x|y,a) = 0$ for $x \neq y$. Also, by Kolmogorov backward equation,

$$\frac{\partial}{\partial t} q_{T-\tilde{t}|T-t,T-s}(x|y,a) = -\frac{\partial}{\partial (T-t)} q_{T-\tilde{t}|T-t,T-s}(x|y,a)$$

$$= \sum_{\tilde{x} \in [S]^d} q_{T-\tilde{t}|T-t,T-s}(\tilde{x}|y,a) R_{T-t|T-s}(\tilde{x},x|a),$$

where $R_{T-t|T-s}(\cdot,\cdot|a)$ is the rate matrix for the forward CTMC at time $T-t$ conditioned on the future event that the observation at time $T-s$ is $a$. Then, combining both sides, we would get

$$\overleftarrow{R}_t(x,y) = \frac{\overleftarrow{q}_{t|s}(y|a)}{\overleftarrow{q}_{t|s}(x|a)} R_{T-t|T-s}(y,x|a).$$

Lastly, we ensure that the conditioned rate is well-defined. Fix $s' < t' < T'$. Note that

$$R_{t'|T'}(y,x|a) = \lim_{s' \to t'} \frac{\partial}{\partial t'} q_{t'|s',T'}(x|y,a).$$

Thus, we obviously have $R_{t'|T'}(y,x|a) = 0$ for all $x \neq y$ if $q_{T'|t'}(a|y) = 0$. Otherwise, we have

$R_{t'|T'}(y,x|a)$

$$= \lim_{s' \to t'} \frac{\partial}{\partial t'} q_{t'|s',T'}(x|y,a)$$

$$= \lim_{s' \to t'} \frac{1}{q_{T'|s'}(a|y)} \frac{\partial}{\partial t'} \left( q_{T'|s',t'}(a|y,x) q_{t'|s'}(x|y) \right)$$

$$= \lim_{s' \to t'} \frac{1}{q_{T'|s'}(a|y)} \frac{\partial}{\partial t'} \left( q_{T'|t'}(a|x) q_{t'|s'}(x|y) \right)$$

$$= \lim_{s' \to t'} \frac{1}{q_{T'|s'}(a|y)} \left( -q_{t'|s'}(x|y) \sum_{\tilde{x} \in [S]^d} R_{t'}(x,\tilde{x}) q_{T'|t'}(a|\tilde{x}) + q_{T'|t'}(a|x) \sum_{\tilde{x} \in [S]^d} q_{t'|s'}(\tilde{x}|y) R_{t'}(\tilde{x},x) \right)$$

$$= \frac{q_{T'|t'}(a|x)}{q_{T'|t'}(a|y)} R_{t'}(y,x)$$

which is finite and non-negative when $x \neq y$. The proof is now complete. $\square$

