# OpenReview forum: "Discrete Diffusion Models: Novel Analysis and New Sampler Guarantees"
_NeurIPS.cc/2025/Conference — NeurIPS 2025 poster_

### Official Review · Reviewer_Hz15 · 2025-06-16

**Clarity:** 3
**Significance:** 2
**Originality:** 3
**Rating:** 5
**Confidence:** 2

**Summary:**

This paper introduces a new analytical approach to analyze samplers for the backward process in discrete-state continuous-time diffusion models. With this new approach, the authors report convergence guarantees for the tau-leaping sampler in KL divergence that scale linearly with the state space size per dimension, in contrast to previous results that showed a quadratic dependence.
Additionally their method can be applied to the euler and tweedie tau leaping sampler, which are defined on a discrete sampling grid,
providing first convergence guarantees.

**Questions:**

Can these results point researchers in the right direction when choosing a sampler for their diffusion model?

**Ethical Concerns:**

["NO or VERY MINOR ethics concerns only"]

**Final Justification:**

I appreciate the author’s responses to my questions. Based on the clarifications provided, I will raise my score.

**Limitations:**

Yes

**Quality:**

3

**Strengths And Weaknesses:**

This paper is an important theoretical contribution to the field of discrete diffusion models in continuous-time, providing valuable convergence guarantees that are valuable for the community.

The paper is well written, providing a good overview over the deterministic-step-size samplers used within the discrete diffusion models and the prior theoretical work on the convergence of the tau-leaping sampler.

The main strength of the paper is the proposed novel analysis based on the rate-of-change of the KL between posterior and sampling distribution as an alternative to the analysis based on the Girsanov change-of-measure framework. This brings two benefits: First, it allows application to samplers such as the Euler and tweedie tau-leaping for which the posterior rates are not defined on the whole continuous-time path. Second, the new method does not require regularity conditions.

Another strength of the paper are the convergence results, which either improve prior results or provide new guarantees.

One weakness of the paper is a missing discussion how this new results can be interpreted for practitioners, when choosing the sampler for their diffusion model.

Typo
Line 76 practivally

---

> ### Author Rebuttal · Authors · 2025-07-30
>
> We sincerely thank the reviewer for providing the highly valuable feedback. We greatly appreciate your time and effort in reviewing our paper.
>
>
> **Q1:** One weakness of the paper is a missing discussion of how these new results can be interpreted for practitioners, when choosing the sampler for their diffusion model.
>
> **A1:** We thank the reviewer for pointing out this important issue. Yes — one key motivation for our work is to inform the design and selection of samplers for discrete diffusion models based on theoretical guarantees. Our results suggest that Euler and Tweedie $\tau$-leaping samplers offer the same convergence rate in KL-divergence as vanilla $\tau$-leaping (Theorem 3), while requiring **lower per-step sampling complexity** by a factor of $O(S)$, where $S$ is the vocabulary size. This makes the former two samplers more scalable for large-vocabulary tasks. Overall, our findings indicate that Euler or Tweedie $\tau$-leaping may be preferable in practice — especially in settings where sampling efficiency is critical and the vocabulary size $S$ is large.
>
> We also provided numerical experiments to validate our theoretical results and to support their practical implications discussed above. Table 1 demonstrates an approximately linear relationship between the estimated KL divergence and the vocabulary size $S$, validating our theoretical prediction of linear dependence on $S$. Table 2 compares the performance of the Euler method and Tweedie $\tau$-leaping against the standard $\tau$-leaping algorithm, highlighting their superior efficiency and practical advantage.
>
>
>
> | Vocabulary Size ($S$) |Estimated KL Divergence |
> | - | - |
> | 3 | 5.93$\times$$10^{-3}$ |
> | 4 | 1.32$\times$$10^{-2}$ |
> | 5 | 2.10$\times$$10^{-2}$ |
> | 6 | 3.38$\times$$10^{-2}$ |
> | 7 | 3.77$\times$$10^{-2}$ |
>
>
> Table 1: Estimated KL-divergence between the target and the sampling distribution. The target is generated autoregressively over the dimensions. Here $d = 2$. We use the Euler method to obtain samples.
>
> | Number of steps | 16 | 32 | 64 | 128 |
> | - | - | - | - | - |
> | Vanilla $\tau$-leaping | 0.290 | 0.215 | 0.179 | 0.161 |
> | Euler method | 0.193 | 0.165 | 0.152 | 0.151 |
> | Tweedie $\tau$-leaping | 0.176 | 0.158 | 0.155 | 0.153 |
>
> Table 2: Estimated total variation distance between the target and sampling distribution of different sampling methods. The same target is used, with $d=3$ and $S=8$.
>
> **Q2:** Typo Line 76 practically
>
> **A2:** Thank you for catching this typo! We will correct it in the revision.
>
> **Q3:** Can these results point researchers in the right direction when choosing a sampler for their diffusion model?
>
> **A3:** Please see our detailed response in A1. We believe our results suggest that Euler or Tweedie $\tau$-leaping may be more favorable in practice, particularly in scenarios where sampling efficiency is crucial and the vocabulary size $S$ is large.
>
> We sincerely thank the reviewer again for the thoughtful comments. We hope that our responses have resolved your concerns. Certainly, we are more than happy to answer any further questions you may have.

---

> ### Comment · Reviewer_Hz15 · 2025-08-01
>
> I appreciate the author’s responses to my questions. Based on the clarifications provided, I will raise my score.

---

> > ### Author Response · Authors · 2025-08-01
> >
> > We sincerely thank the reviewer for the prompt feedback and for kindly raising the rating.

---

### Official Review · Reviewer_EBtu · 2025-07-01

**Clarity:** 3
**Significance:** 4
**Originality:** 3
**Rating:** 5
**Confidence:** 3

**Summary:**

Thanks to suthors for submitting to NeuRIPs 2025. Their work addresses theoretical aspects of discrete diffusion models and derives novels framework for convergence bounds. Paper addresses two challenges present in current convergence bounds for $\tau$-leaping sampler and its variants: 1. hard-to-verify assumptions (l 53-59) and 2. Better scaling - in terms of vocabulary size as well as for $\tau$-leaping variants with deterministic step size, i.e., Euler and Tweedie, used in practise.
By carefully treating the discrete nature of the problem, the Kolmogorov forward eq. is used and upper bounded by bounding the leading term to replace a need of It\^o calculus and Girsanov change of measure technique. Insuch doing, for specific parameterisation (5) the main result (Theorem 2) proves the linearly scaling bound in vocabulary size for $\tau$-leaping sampler and its variants, improving existing quadratic bounds for $\tau$-leaping and introducing novel results for Euler  and Tweedie.

**Questions:**

Q1: Wider discussion or experiments concerning limitations/ablation study/asymptotics along the lines of Weakness W1 is suggested.

Q2: See weakness W2.

Q3: Could authors elaborate why $g_t$ is a Bregman divergence, whose non-negativity was used in proof (Appendix F)? Bregman div. is defined by $\phi(y) - \phi(x) - \langle y-x, \nabla\phi(y)\rangle$, where $\phi$ is strictly convex function (here negative entropy), if I am not mistaken. Could authors rewrite presented relation to this form (or show it otherwise please?)

**Ethical Concerns:**

["NO or VERY MINOR ethics concerns only"]

**Final Justification:**

Thanks to Authors for carefully addressing questions raised, especially Q1-Q2. I do not have any further comments and find paper to be novel and valuable for ML community and recommend it to be accepted (keep my current 'Accept' score).

**Limitations:**

See W1

**Paper Formatting Concerns:**

Paper follows pure theory structure, i.e., main body is filled by proof ideas and no experiments and (wider) discussion is given. Leave it for AC to decide on suitability for NeuRIPs conference. Fine by me.

**Quality:**

3

**Strengths And Weaknesses:**

Well motivated and timely topic. Introduction is well versed and provide context to problem addressed by the paper.

The paper is theoretical in nature and provides a rigorous treatment of the topic, referencing existing results and sources of inspiration (e.g., l 223) while emphasising the non-trivial novelties introduced by authors. So despite quite technical topic, the approach and assumptions needed are relatively straightforward to follow.

Contributions are both theoretical (novel theoretical results) as well as practical (new guarantess for popular algorithms) and thus paper is well suited for wider ML community.

On the improvemnt side:
W1: Limitations: Presented limitations and wider discussion is very limited and perhaps too slim. For instance, the main results are derived for special CTMC rate matrix (5). To what extend is this limiting? If it is, it should be emphasized. Can it be extended? The same goes for Assumtions 1 and 2. One of the claims of the paper is it alleviates difficulties with hard-to-check assumptions. It would be worth discussing or ortherwise showing (experiments?) the new assumtions are easier to verify/meet.

W2: Experiments/Ablation study. Despite its theoretical nature supporting experiment(s) could increase accessability and appeal fo the paper. For instance, Main result (Th 2) is an asymptotic result (in T) in its nature. Could authors show in the experiment (or otherwise) for what ranges of T asymptotic behaviour prevails? Is it data/task dependent?

Minor:
- numbering of equations is missing as of Section 3, included.
- (l.140) define $\mathbb{1}{y=x}$ and other symbiolics used

---

> ### Author Rebuttal · Authors · 2025-07-30
>
> We sincerely thank the reviewer for providing valuable feedback and greatly appreciate your time and effort in reviewing our paper.
>
> **Q1:** Presented limitations and wider discussion is very limited and perhaps too slim. For instance, the main results are derived for a special CTMC rate matrix (5). To what extent is this limiting? If it is, it should be emphasized. Can it be extended?
>
> **A1:** We thank the reviewer for pointing this out. We agree that our latter theoretical results (namely Theorems 2 and 3) are derived under the commonly used rate estimator in (5), which ensures analytical tractability and aligns with prior work (e.g., [1-3]). However, our analytical framework — particularly Theorem 1 — is **not limited to this specific reverse rate parameterization** and applies to any CTMC as long as the reverse rate matrix is well-defined. Meanwhile, although we assume a particular parameterization for the reverse rate in Theorems 2 and 3, they can be easily extended for general reverse rates with slightly modified Assumptions 1 and 2 (which are then made on the estimated rates instead of the scores). We will discuss our current limitation on the uniform rate matrix under time-homogeneity and potential extensions to other rate matrices and under non-homogeneous noise levels in the revision. We will further discuss another limitation on using Bergman divergence to characterize the estimation error. It might be interesting to investigate other forms of estimation error, such as in $L_1$ or $L_2$.
>
> **Q2:** The same goes for Assumptions 1 and 2. One of the claims of the paper is it alleviates difficulties with hard-to-check assumptions. It would be worth discussing or otherwise showing (experiments?) the new assumptions are easier to verify/meet.
>
> **A2:** We appreciate the reviewer’s thoughtful comment. Both Assumptions 1 and 2 are easy to check/enforce. Assumption 2 can be easily enforced through score-clipping. In particular, during training we can clip those scores that are too small or large with a constant $M$ [4]. Also, note that our convergence result only depends on $\log M$. Assumption 1 requires bounding the score-entropy loss evaluated on the discretized grid, which is easy to check. In particular, we can take the average loss (up to some constant) on the selected grid. This is in contrast to those integrated loss functions [4, 5], where taking an integral is needed to evaluate the loss.
>
> Just to clarify as a side note, both Assumptions 1 and 2 also occur in prior works (e.g., [3-4]).
>
> **Q3:** Experiments/Ablation study. Despite its theoretical nature, supporting experiment(s) could increase accessibility and appeal for the paper. For instance, Main result (Th 2) is an asymptotic result (in T) in its nature. Could authors show in the experiment (or otherwise) for what ranges of T asymptotic behaviour prevails? Is it data/task dependent?
>
> **A3:** To clarify, the asymptotic results are in terms of the final error $\epsilon$, which is affected, for example, by the number of steps chosen. We have included two tables to provide empirical support for our theoretical results. Table 1 demonstrates an approximately linear relationship between the estimated KL divergence and the vocabulary size $S$, validating our theoretical prediction of linear dependence on $S$, even for moderate number of steps (instead of asymptotically large ones). Please see our response to reviewer 1 for more details.
>
> Table 2 compares the performance of the Euler method and Tweedie $\tau$-leaping against the standard $\tau$-leaping algorithm, highlighting their superior efficiency and practical advantage.
>
>
> | Vocabulary Size ($S$) |Estimated KL Divergence |
> | - | - |
> | 3 | 5.93$\times$$10^{-3}$ |
> | 4 | 1.32$\times$$10^{-2}$ |
> | 5 | 2.10$\times$$10^{-2}$ |
> | 6 | 3.38$\times$$10^{-2}$ |
> | 7 | 3.77$\times$$10^{-2}$ |
>
>
> Table 1: Estimated KL-divergence between the target and the sampling distribution. The target is generated autoregressively over the dimensions. Here $d = 2$. We use the Euler method to obtain samples.
>
> | Number of steps | 16 | 32 | 64 | 128 |
> | - | - | - | - | - |
> | Vanilla $\tau$-leaping | 0.290 | 0.215 | 0.179 | 0.161 |
> | Euler method | 0.193 | 0.165 | 0.152 | 0.151 |
> | Tweedie $\tau$-leaping | 0.176 | 0.158 | 0.155 | 0.153 |
>
> Table 2: Estimated total variation distance between the target and sampling distribution of different sampling methods. The same target is used, with $d=3$ and $S=8$.
>
> In general, asymptotic results usually have a vanishing constant, which depends on other system parameters such as the data dimension, yielding the tightness of these theoretical bounds data/task dependent. Yet, our experiment above shows that such asymptotic behavior might occur very early (i.e., when the number of steps are still moderate). Also, our main result in Theorem 3 further shows the advantage of Euler and Tweedie $\tau$-leaping samplers, both of which are commonly used in practice.
>
> **Q4:** Minor and typos
>
> **A4:** We thank the reviewer for pointing these out. We will add the corresponding definitions and line numbers, and correct typos in the revised paper.
>
> **Q5:** Could authors elaborate why $g_t$ is a Bregman divergence, whose non-negativity was used in proof (Appendix F)? Bregman div. is defined by $\phi(y)-\phi(x)-<y-x,\phi(y)>$, where $\phi$ is a strictly convex function (here negative entropy), if I am not mistaken. Could authors rewrite the presented relation to this form (or show it otherwise please?)
>
> **A5:** Definitely. In our case, for each fixed $x_t$ and $y \neq x_t$, let $$p_y := R_t(x_t,y), q_y := \\hat{R}_t(x_t,y).$$
>
> Also, let $\phi(p) := \sum_{y \neq x_t} p_y \log p_y$ (i.e., the negative entropy function, which is convex), and the corresponding Bregman divergence is $D_\phi (p, q) = \phi(p) - \phi(q) - \langle y-x,\phi(y) \rangle = \sum_{y \neq x_t} p_y \log p_y - q_y \log q_y - (p_y - q_y) (1 + \log q_y) = \sum_{y \neq x_t} p_y \log(p_y / q_y) - p_y + q_y,$ which is exactly $g_t$. We will add this part in Appendix F for clarification.
>
> We sincerely thank the reviewer again for the thoughtful comments. We hope that our responses have resolved your concerns.  Certainly, we are more than happy to answer your further questions if any.
>
> [1] Campbell, Andrew, et al. “A Continuous Time Framework for Discrete Denoising Models.”
>
> [2] Lou, Aaron, et al. “Discrete diffusion modeling by estimating the ratios of the data distribution.”
>
> [3] Ren, Yinuo, et al. “How Discrete and Continuous Diffusion Meet: Comprehensive Analysis of Discrete Diffusion Models via a Stochastic Integral Framework.”
>
> [4] Zhang, Zikun, et al. “Convergence of Score-Based Discrete Diffusion Models: A Discrete-Time Analysis.”
>
> [5] Chen, Hongrui and Ying, Lexing. “Convergence analysis of discrete diffusion model: Exact implementation through uniformization.”
>
> [6] Pham, Le-Tuyet-Nhi et al. “Discrete markov probabilistic models.”

---

> ### Comment · Area_Chair_85ox · 2025-08-04
> **Subject: Friendly Reminder to Acknowledge or Update Your Review**
>
> Dear Reviewer EBtu,
>
> Thank you for your time and effort in reviewing the submissions and for providing valuable feedback to the authors.
>
> If you haven’t already done so, we kindly remind you to review the authors’ rebuttals and respond accordingly. In particular, if your evaluation of the paper has changed, please update your review and explain the revision. If not, we would appreciate it if you could acknowledge the rebuttal by clicking the “Rebuttal Acknowledgement” button at your earliest convenience.
>
> This step ensures smooth communication and helps us move forward efficiently with the review process.
>
> We sincerely appreciate your dedication and collaboration.
>
> Best regards,
> AC

---

### Official Review · Reviewer_ZqTY · 2025-07-03

**Clarity:** 3
**Significance:** 4
**Originality:** 3
**Rating:** 5
**Confidence:** 3

**Summary:**

This paper presents a novel theoretical tool for convergence analysis of discrete diffusion models.
The approach is based on differential-inequality, which circumvents the need for the regularity assumptions required by Girsanov theorem.

Theoretically, this new framework provides an improved convergence guarantee for $\tau$-leaping sampler.
The linear dependence on the vocabulary size $S$ improves upon the previously
results $\mathcal{O}(S^2)$ complexity.
Moreover, the authors provides the convergence order for the Euler method and Tweedie $\tau$-leaping samplers.
These two sampling schemes are shown to have the same convergence rate in terms of the number of steps, but with a lower per-step sampling complexity.

**Questions:**

Please see the Weaknesses section.

**Ethical Concerns:**

["NO or VERY MINOR ethics concerns only"]

**Final Justification:**

After reviewing the authors' response to my question, I am now convinced of the manuscript's contribution and recommend its acceptance.

**Limitations:**

The main text lacks a discussion of the limitations.

**Quality:**

3

**Strengths And Weaknesses:**

### Strength
- The paper's contribution is clear: it provides an improved convergence bound for the $\tau$-leaping sampler using a novel analytical technique.  Notably, this approach also weakens standard regularity assumptions in Girsanov theorem without introducing any new ones.
- The paper is well-written and provides a dedicated section for its key notations (Section 2.4).


### Weaknesses
The theoretical analysis is strong. One question for the authors is whether they considered numerical validation for the convergence rate with respect to vocabulary size $S$.

---

> ### Author Rebuttal · Authors · 2025-07-30
>
> We sincerely thank the reviewer for providing valuable feedback and greatly appreciate your time and effort in reviewing our paper.
>
> **Q1:** One question for the authors is whether they considered numerical validation for the convergence rate with respect to vocabulary size $S$.
>
> **A1:** Thank you for raising this point! We have included two tables to provide empirical support for our theoretical results. Table 1 demonstrates an approximately linear relationship between the estimated KL divergence and the vocabulary size $S$, validating our theoretical prediction of linear dependence on $S$. Table 2 compares the performance of the Euler method and Tweedie $\tau$-leaping against the standard $\tau$-leaping algorithm, highlighting their superior efficiency and practical advantage.
>
> | Vocabulary Size ($S$) |Estimated KL Divergence |
> | - | - |
> | 3 | 5.93$\times$$10^{-3}$ |
> | 4 | 1.32$\times$$10^{-2}$ |
> | 5 | 2.10$\times$$10^{-2}$ |
> | 6 | 3.38$\times$$10^{-2}$ |
> | 7 | 3.77$\times$$10^{-2}$ |
>
>
> Table 1: Estimated KL-divergence between the target and the sampling distribution. The target is generated autoregressively over the dimensions. Here $d = 2$. We use the Euler method to obtain samples.
>
> | Number of steps | 16 | 32 | 64 | 128 |
> | - | - | - | - | - |
> | Vanilla $\tau$-leaping | 0.290 | 0.215 | 0.179 | 0.161 |
> | Euler method | 0.193 | 0.165 | 0.152 | 0.151 |
> | Tweedie $\tau$-leaping | 0.176 | 0.158 | 0.155 | 0.153 |
>
> Table 2: Estimated total variation distance between the target and sampling distribution of different sampling methods. The same target is used, with $d=3$ and $S=8$.
>
> We sincerely thank the reviewer again for the thoughtful comments. We hope that our responses have resolved your concerns. Certainly, we are more than happy to answer your further questions if any.

---

> ### Comment · Area_Chair_85ox · 2025-08-04
> **Subject: Friendly Reminder to Acknowledge or Update Your Review**
>
> Dear Reviewer ZqTY,
>
> Thank you for your time and effort in reviewing the submissions and for providing valuable feedback to the authors.
>
> If you haven’t already done so, we kindly remind you to review the authors’ rebuttals and respond accordingly. In particular, if your evaluation of the paper has changed, please update your review and explain the revision. If not, we would appreciate it if you could acknowledge the rebuttal by clicking the “Rebuttal Acknowledgement” button at your earliest convenience.
>
> This step ensures smooth communication and helps us move forward efficiently with the review process.
>
> We sincerely appreciate your dedication and collaboration.
>
> Best regards,
> AC

---

### Official Review · Reviewer_qy7m · 2025-07-07

**Clarity:** 4
**Significance:** 3
**Originality:** 3
**Rating:** 4
**Confidence:** 4

**Summary:**

The paper introduces a novel analytical framework for discrete diffusion models based on differential inequalities. This new technique circumvents the need for restrictive regularity assumptions previously required by methods relying on the Girsanov change-of-measure framework. The primary contributions are: (1) The development of a more flexible analysis tool for discrete diffusion models. (2) An improved convergence guarantee for the popular $\tau$-leaping sampler, as well as the first-ever convergence guarantees for the empirically efficient Euler method and Tweedie $\tau$-leaping samplers.

**Questions:**

- Your results in Theorems 2 and 3 show that the Euler and Tweedie $\tau$-leaping samplers have the same asympotic convergence rate as $\tau$-leaping, but with a significantly lower per-step sampling complexity. Does it suggest that there are larger hidden constants in the convergence bounds for these samplers that may impact their practical performance?
- Can your analysis extend to the high-order samplers as in [1]?

[1] Ren, Yinuo, et al. "Fast solvers for discrete diffusion models: Theory and applications of high-order algorithms." arXiv preprint arXiv:2502.00234 (2025).

**Ethical Concerns:**

["NO or VERY MINOR ethics concerns only"]

**Final Justification:**

I am keeping my rating (4, Borderline Accept), as this paper is well-written and provides improved results based on previous works.

**Quality:**

3

**Strengths And Weaknesses:**

### Strength
- The novel differential inequality analysis removes restrictive Girsanov-based regularity assumptions, a significant theoretical advance.
- The improved linear dependency on vocabulary size S for $\tau$-leaping is a key result with practical importance.
- This paper provides the first-ever convergence guarantees for the efficient Euler and Tweedie $\tau$-leaping samplers.
- The paper is clear and well-organized, and the included comparison table is very helpful for context.

### Weaknesses
- The work is entirely theoretical and lacks any empirical validation to support its findings.
- The convergence rates maintain a quadratic dependence on data dimension $d$, which remains a practical limitation.

---

> ### Author Rebuttal · Authors · 2025-07-30
>
> We sincerely thank the reviewer for providing valuable feedback and greatly appreciate your time and effort in reviewing our paper.
>
> **Q1:** The work is entirely theoretical and lacks any empirical validation to support its findings.
>
> **A1:** We appreciate the reviewer’s concern. We have included two tables to provide empirical support for our theoretical results. Table 1 demonstrates an approximately linear relationship between the estimated KL divergence and the vocabulary size $S$, validating our theoretical characterization of linear dependence on $S$. Table 2 compares the performance of the Euler method and Tweedie $\tau$-leaping against the standard $\tau$-leaping algorithm (in terms of the estimated TV distance between the target and the sampling distribution), highlighting their superior efficiency and practical advantage.
>
> | Vocabulary Size ($S$) |Estimated KL Divergence |
> | - | - |
> | 3 | 5.93$\times$$10^{-3}$ |
> | 4 | 1.32$\times$$10^{-2}$ |
> | 5 | 2.10$\times$$10^{-2}$ |
> | 6 | 3.38$\times$$10^{-2}$ |
> | 7 | 3.77$\times$$10^{-2}$ |
>
>
> Table 1: Estimated KL-divergence between the target and the sampling distribution. The target is generated autoregressively over the dimensions. Here $d = 2$. We use the Euler method to obtain samples.
>
> | Number of steps | 16 | 32 | 64 | 128 |
> | - | - | - | - | - |
> | Vanilla $\tau$-leaping | 0.290 | 0.215 | 0.179 | 0.161 |
> | Euler method | 0.193 | 0.165 | 0.152 | 0.151 |
> | Tweedie $\tau$-leaping | 0.176 | 0.158 | 0.155 | 0.153 |
>
> Table 2: Estimated total variation distance between the target and sampling distribution of different sampling methods. The same target is used, with $d=3$ and $S=8$.
>
> Please note that the empirical performance of these practical samplers like $\tau$-leaping, Euler, and Tweedie $\tau$-leaping have also been widely studied in many existing experimental works, e.g., [1-3].
>
> **Q2:** The convergence rates maintain a quadratic dependence on data dimension $d$, which remains a practical limitation.
>
> **A2:** We thank the reviewer for this thoughtful observation. Indeed, our convergence bounds scale quadratically in $d$, which is consistent with the prior work of [4]. While reducing this dependency remains an important open direction, our main contribution is the **reduction of the dependence on vocabulary size $S$** from quadratic to linear — a critical improvement in NLP and sequence modeling where $S \gg d$. For example, in [1-2], $S$ is as high as 50257 while $d$ is only 1024.
>
> **Q3:** Your results in Theorems 2 and 3 show that the Euler and Tweedie $\tau$-leaping samplers have the same asymptotic convergence rate as $\tau$-leaping, but with a significantly lower per-step sampling complexity. Does it suggest that there are larger hidden constants in the convergence bounds for these samplers that may impact their practical performance?
>
> **A3:** In general, our results suggest $\tau$-leaping has the same convergence rate (in terms of **number of iterations**) as the Euler and Tweedie $\tau$-leaping samplers. Hence, with respect to the number of iterations, our results do not suggest large hidden constants in the convergence bounds. In particular, Lemma 7 establishes the asymptotic equivalency between Euler, Tweedie $\tau$-leaping, and the truncated $\tau$-leaping algorithms, where the last one is itself a rate-truncated version of the standard $\tau$-leaping algorithm. Thus, the convergence rate of Euler and Tweedie $\tau$-leaping samplers should be the same as vanilla $\tau$-leaping, without any hidden constants.
>
> However, in practice, performance often corresponds more closely to the **total number of samples taken**—that is, the product of the number of samples per step and the total number of iterations. In this context, the Euler and Tweedie $\tau$-leaping samplers offer advantages over the vanilla $\tau$-leaping method due to their lower per-step sampling cost. In particular, new experimental result seems to suggest that these samplers differ in their constant-level performance, with the Euler method and Tweedie $\tau$-leaping exhibiting **consistently smaller** constants than vanilla $\tau$-leaping, likely due to their more efficient sampling in each iteration step.
>
> **Q4:** Can your analysis extend to the high-order samplers as in [5]?
>
> **A4:** The core idea of [5] is to better approximate the “middle” rate through a Runge-Kutta-like method designed for CTMC. With such a “middle” rate, [5] shows an accelerated convergence result (in terms of $\epsilon$) for the $\tau$-leaping algorithm. At a high level, it appears that our proof framework can be extended to [5] to investigate the “middle” rate for the truncated version of the vanilla $\tau$-leaping reverse rate and obtain an accelerated convergence result for Euler and Tweedie $\tau$-leaping samplers. However, this extension will involve the careful development of some detailed technical steps and would require a rigorous analysis.
>
> We sincerely thank the reviewer again for the thoughtful comments. We hope that our responses have resolved your concerns. Certainly, we are more than happy to answer your further questions if any.
>
> [1] Lou, Aaron, et al. “Discrete diffusion modeling by estimating the ratios of the data distribution.”
>
> [2] Ou, Jingyang, et al. “Your Absorbing Discrete Diffusion Secretly Models the Conditional Distributions of Clean Data.”
>
> [3] Nisonoff, Hunter, et al. “Unlocking Guidance for Discrete State-Space Diffusion and Flow Models.”
>
> [4] Ren, Yinuo, et al. “How Discrete and Continuous Diffusion Meet: Comprehensive Analysis of Discrete Diffusion Models via a Stochastic Integral Framework.”
>
> [5] Ren, Yinuo, et al. "Fast solvers for discrete diffusion models: Theory and applications of high-order algorithms."

---

> > ### Comment · Reviewer_qy7m · 2025-08-05
> >
> > I am keeping my rating (4, Borderline Accept), as this paper is well-written and provides improved results based on previous works.

---

> ### Comment · Area_Chair_85ox · 2025-08-04
> **Subject: Friendly Reminder to Acknowledge or Update Your Review**
>
> Dear Reviewer qy7m,
>
> Thank you for your time and effort in reviewing the submissions and for providing valuable feedback to the authors.
>
> If you haven’t already done so, we kindly remind you to review the authors’ rebuttals and respond accordingly. In particular, if your evaluation of the paper has changed, please update your review and explain the revision. If not, we would appreciate it if you could acknowledge the rebuttal by clicking the “Rebuttal Acknowledgement” button at your earliest convenience.
>
> This step ensures smooth communication and helps us move forward efficiently with the review process.
>
> We sincerely appreciate your dedication and collaboration.
>
> Best regards,
> AC

---

### Official Review · Reviewer_ieBG · 2025-07-09

**Clarity:** 3
**Significance:** 3
**Originality:** 3
**Rating:** 5
**Confidence:** 2

**Summary:**

This paper extends the theoretical guarantee if discrete diffusion samplers with a novel framework. Specifically, the quadratic dependency on vocabulary size in the iteration guarantee is improved to linear and new convergence on two practically efficient samplers are further provided.

**Questions:**

I don't have further questions.

**Ethical Concerns:**

["NO or VERY MINOR ethics concerns only"]

**Final Justification:**

After reading the other valuable reviews, I believe that the quality and potential impact of this paper are worthy of recognition. I keep my current rating.

**Limitations:**

yes

**Paper Formatting Concerns:**

I don't find any major formatting issues in this paper.

**Quality:**

3

**Strengths And Weaknesses:**

Strengths:
- The paper is well-structured and well-written with thorough proof. Assumptions are properly discussed. Detailed discussion and a comprehensive comparison with related works are included, which is very helpful.
- The theoretical contribution is clear and strong. The convergence guarantee is proved without regularity conditions. The iteration guarantee of $\tau$-leaping is improved and new convergence results on practical samplers are provided.

I am satisfied with this paper and I don't find any significant issues. I am familiar with diffusion models but not particularly familiar with this specific field. I will continue to refer to the opinions of other reviewers for a more comprehensive evaluation.

---

> ### Author Rebuttal · Authors · 2025-07-30
>
> We sincerely thank the reviewer for providing valuable feedback, for the positive comments,  and for recognizing our contributions. We greatly appreciate your time and effort in reviewing our paper.

---

> ### Comment · Area_Chair_85ox · 2025-08-04
> **Subject: Friendly Reminder to Acknowledge or Update Your Review**
>
> Dear Reviewer ieBG,
>
> Thank you for your time and effort in reviewing the submissions and for providing valuable feedback to the authors.
>
> If you haven’t already done so, we kindly remind you to review the authors’ rebuttals and respond accordingly. In particular, if your evaluation of the paper has changed, please update your review and explain the revision. If not, we would appreciate it if you could acknowledge the rebuttal by clicking the “Rebuttal Acknowledgement” button at your earliest convenience.
>
> This step ensures smooth communication and helps us move forward efficiently with the review process.
>
> We sincerely appreciate your dedication and collaboration.
>
> Best regards,
> AC

---

### Decision · Program_Chairs · 2025-09-17

**Decision:**

Accept (poster)

**Comment:**

This work studies the sample complexity of discrete diffusion models (for popular sampling algorithms such as $\tau$-leaping, Euler method, and Tweedie $\tau$-leaping algorithm) and improves the dependence of vocabulary size $S$ under weaker assumptions compared with previous works. To achieve the guarantee under the KL divergence, this work adopts the Kolmogorov equation instead of the Girsanov change-of-measure framework. Finally, they do a detailed analysis on the CTMC rate matrix to remove the dependence of $S$.

All reviewers keep a positive score for this work and view this work as a solid step in the analysis of discrete diffusion models. In the rebuttal phase, this work provide the additional simulation experiments to support the theoretical results to address the concerns of the reviewers, and the concerns of reviewers are addressed. Hence, we make a accept recommendation.